# LoTLIP: Improving Language-Image Pre-training for Long Text Understanding

**Wei Wu**[1]  **Kecheng Zheng**[2,3†]  **Shuailei Ma**[4]  **Fan Lu**[1]  **Yuxin Guo**[5]  **Yifei Zhang**[6]
**Wei Chen**[3]  **Qingpei Guo**[2]  **Yujun Shen**[2]  **Zheng-Jun Zha**[1†]

[1]University of Science and Technology of China   [2]Ant Group   [3]Zhejiang University
[4]Northeastern University, China   [5]Institute of Automation, Chinese Academy of Sciences
[6]Shanghai Jiao Tong University

## Abstract

Understanding long text is of great demands in practice but beyond the reach of most language-image pre-training (LIP) models. In this work, we empirically confirm that the key reason causing such an issue is that the training images are usually paired with short captions, leaving certain tokens easily overshadowed by salient tokens. Towards this problem, our initial attempt is to relabel the data with *long captions*, however, directly learning with which may lead to performance degradation in understanding short text (*e.g.*, in the image classification task). Then, after incorporating corner tokens to aggregate diverse textual information, we manage to help the model catch up to its original level of short text understanding yet greatly enhance its capability of long text understanding. We further look into whether the model can continuously benefit from longer captions and notice a clear trade-off between the performance and the efficiency. Finally, we validate the effectiveness of our approach using a self-constructed large-scale dataset, which consists of $100M$ long caption oriented text-image pairs. Our method achieves superior performance in long-text-image retrieval tasks. The project page is available here.

## 1   Introduction

Understanding long texts plays a key role in Natural Language Processing (NLP), *e.g.*, Long Document Analysis [23, 28, 33], in which books, academic papers, and many other types of long texts can be the target of such analysis. Inspired by these works, in the multi-modality field, some text-to-image generation works (*e.g.*, DALLE-3 [1] and Pixel-art [4, 5]) employ pre-trained captioners (*e.g.*, LLaVA [21]) to generate more accurate and detailed captions (in other words, long captions) for images. These long captions describe an image in detail which can help text-to-image models easily transfer long text to an image, improving the quality of text-image alignment. However, these text-to-image models use pure NLP encoders (*e.g.*, T5) to model long captions rather than CLIP. Because, in the language-image pretraining task, little work has been conducted on modeling long texts in a way that effectively aligns with image representations. Despite the lack of exploration, this is a critical problem with practical demands that require a nuanced understanding of textual descriptions corresponding to images.

Two primary challenges hinder the effective integration of long-text understanding in language-image pre-training. The first one is the lack of large-scale long-caption image-text paired datasets. Most

---

† Corresponding authors.

38th Conference on Neural Information Processing Systems (NeurIPS 2024).

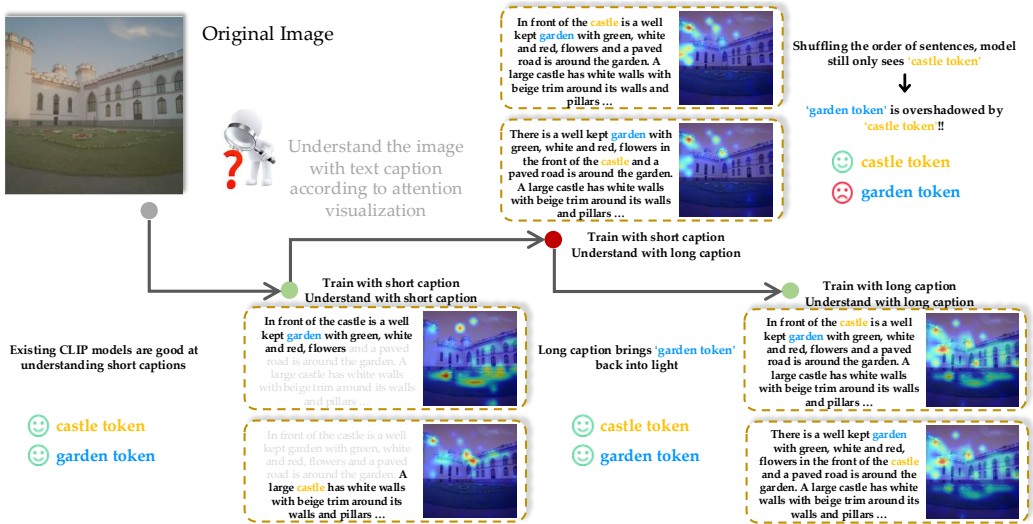

Figure 1: Illustration of the impacts of long *v.s.* short captions on image-language pre-training, as observed in the cross-attention maps of CLIP. Training images are usually paired with short captions, leaving certain tokens (*e.g.*, garden token) easily overshadowed by salient tokens (*e.g.*, castle token). Fortunately, the usage of long captions can help bring the overshadowed tokens back into the light, and this phenomenon is not influenced by the order of tokens within the sentence.

existing datasets focus on short captions (*i.e.*, average text length in CC12M [3] is about 17 tokens), which limits the model's exposure to longer text forms. Consequently, models trained on these datasets tend to neglect certain tokens that are easily overshadowed by salient tokens. As shown in Fig. 1, the model trained on short captions can be good at understanding the content of short captions (*i.e.*, garden and castle). But when increasing the length of the caption, we can see that 'garden token' is overshadowed by 'castle token', even if we move the 'garden token' front. This bias towards the salient tokens ('castle token' in Fig. 1) of texts can severely restrict the model's ability to comprehend and generate responses based on the full context of longer inputs. The second challenge is the token length limitation of the text encoder. While directly increasing the token number limitation that a model can process appears to be a straightforward solution for accommodating longer texts, it does not have a better understanding of long captions. The fundamental issue remains the model's inability to effectively interpret long texts, primarily due to the lack of appropriate training data that includes long captions.

To address these challenges, we have undertaken an extensive project to re-caption 100 million data with long captions, aiming to enrich the training environment for our models. This initiative allows us to explore the effects of increased text length in image-text pre-trained models and its impacts on model performance. Based on these experiments, we empirically confirm that the key reason causing such an issue is that the training images are usually paired with short captions, leaving certain tokens easily overshadowed by salient tokens. However, directly learning with long captions may improve the long-text understanding of image-text pre-trained models, but lead to performance degradation in understanding short texts (*e.g.*, in the image classification task) as shown in Fig. 2. After integrating corner tokens to aggregate diverse textual information, we successfully enable the model to regain its original proficiency in understanding short texts while significantly improving its ability to comprehend long texts. We also explore whether the model can continue to benefit from longer captions and observe a clear trade-off between performance and efficiency. Moreover, on the task of long-text image retrieval, we beat Long-CLIP [39], a competitor using long captions for fine-tuning, with 4.2% improvement (*i.e.*, from 79.52% to 83.72%).

## 2   Related work

### 2.1   Language-Image Pre-training

Recently, using language-image pre-trained models to do zero-shot prediction has attracted a lot of attention. CLIP [24] and ALIGN [14] demonstrate contrastive pre-trained models can learn

rich visual-language correspondence knowledge from large-scale image-text pairs on the Internet and achieve good performance on zero-shot predictions, including image-text retrieval [18] and classification [9]. Following their success, various studies [16, 34, 36, 17] have been devoted to improving image-text alignment. Among them, FILIP [34] focuses on fine-grained expressiveness between text tokens and image patches by modifying the training loss. While LiT [37] finds that apply contrastive-tuning to the pre-trained models with locked image encoder and unlocked text encoder can further improve the alignment. It is recognized that larger batch size brings better performance. For this reason, SigLIP [38] proposes to replace the softmax normalization among the standard contrastive loss with the sigmoid loss to scale up training batch size. In addition, LaCLIP [11], RLEG [40] and some other works [19, 13, 22, 30] utilize multi-modality generative models to improve data quality for enhancing pre-training.

## 2.2 Long-text Understanding

Detailed long texts are necessary for many artificial intelligence tasks (*e.g.*, human-computer interaction). Therefore, modeling and parsing long texts has become one of the most important research directions in natural language processing. Many studies in fields such as text generation [23, 32, 33] have shown that advanced transformer structures have the ability to interpret long texts. However, in the field of language-image pre-training, research on using long-text descriptions of images to enhance multimodal representations is still very scarce. DreamLIP [41] utilizes multi-modality large language model to re-caption image data with detailed descriptions and then use them in language-image pre-training. In fact, during training, DreamLIP randomly extracts sub-captions from the detailed description for training and does not completely use all the information of the long text. One of the reasons why previous language-image pre-training rarely directly applied long texts for training is the text encoder of traditional CLIP [24] is restricted by the token number limit ($\leq 77$). Then, Long-CLIP [39] firstly introduces long captions into CLIP model to finetune, where the model is pre-trained on short-text-image datasets. The fine-tuning process equips the model with the ability to comprehend long texts, albeit at the expense of its proficiency in understanding shorter texts. In contrast, we incorporate long captions during the pre-training stage, which can not only enable the model to regain its original competency in interpreting short texts but also significantly improves its understanding of long texts. We further explored the trade-off between the benefit of long texts to the model and the training efficiency.

## 3   Preliminary of Language-Image Pre-training

Language-image pre-training models, *i.e.*, CLIP [24] and LiT [37], typically consist of an image encoder and a text encoder. In the pre-training stage, the language-image model takes image-text pairs as input and uses image encoder and text encoder to extract embeddings from images and texts, respectively. Then, two encoders are trained with contrastive objectives, ensuring that paired image and text embeddings are close in the embedding space, while unpaired pairs are far apart. Specifically, a batch of images $\{I_1, I_2, \cdots, I_N\}$ and the corresponding short texts $\{T_1, T_2, \cdots, T_N\}$ are randomly sampled from the pre-training dataset. Each image and its corresponding text are treated as a positive pair, while others in the same batch are negative pairs. Then, the image encoder extracts image global feature $\mathbf{v}^i$ from the $i$-th image $I_i$ within the batch, while the text encoder obtains text feature $\mathbf{t}^j$ from the $j$-th text $T_j$. These two encoders are then optimized using contrastive loss $\mathcal{L}$, which consists of image-to-text loss $\mathcal{L}^{i2t}$ and text-to-image loss $\mathcal{L}^{t2i}$. It can be formulated as follows:

$$\mathcal{L} = \mathcal{L}^{i2t} + \mathcal{L}^{t2i}, \tag{1}$$

$$\mathcal{L}^{i2t} = -\sum_{i=1}^{N} \log \frac{\exp\left(\cos\langle \mathbf{v}^i, \mathbf{t}^i \rangle / \tau\right)}{\sum_{j=1}^{N} \exp\left(\cos\langle \mathbf{v}^i, \mathbf{t}^j \rangle / \tau\right)}, \tag{2}$$

$$\mathcal{L}^{t2i} = -\sum_{i=1}^{N} \log \frac{\exp\left(\cos\langle \mathbf{t}^i, \mathbf{v}^i \rangle / \tau\right)}{\sum_{j=1}^{N} \exp\left(\cos\langle \mathbf{t}^i, \mathbf{v}^j \rangle / \tau\right)}, \tag{3}$$

where $\tau$ is a learnable temperature parameter, and $\cos\langle \cdot, \cdot \rangle$ means the cosine similarity between two normalized feature vectors.

Table 1: **Dataset details of long-text-image retrieval and short-text-image retrieval tasks**. We use BERT tokenizer for tokenization. ShareGPT4V-1k and 10k are selected from the ShareGPT4V dataset. For DCI and IIW, we use images with human-authored long descriptions for evaluation.

| Dataset | #Images | #Texts | #Sub-captions per Text | #Tokens per Text |
|---|---|---|---|---|
| Long-text-image Retrieval Dataset | | | | |
| DCI [31] | 7,805 | 7,805 | 10.81 | 172.73 |
| IIW [12] | 612 | 612 | 10.16 | 239.73 |
| ShareGPT4V-1k [6] | 1,000 | 1,000 | 8.15 | 173.24 |
| ShareGPT4V-10k [6] | 10,000 | 10,000 | 8.24 | 173.66 |
| Short-text-image Retrieval Dataset | | | | |
| MSCOCO [18] | 5,000 | 25,000 | 1.0 | 11.77 |
| Flickr30k [35] | 1,000 | 5,000 | 1.0 | 14.03 |

# 4 Long Texts in Language-Image Pre-training

Currently available image-text pair datasets, *e.g.*CC12M [27], typically include short texts that have an average length of approximately 17 tokens. Language-image models pre-trained on these datasets perform well on short-text comprehension tasks. However, we find that they struggle to comprehend long texts for text-image alignment. Concretely, they tend to overlook or neglect some tokens or sub-captions in long texts, as shown in Fig. 1. A potential reason for such a situation is the lack of long-text-image pairs in pre-training. Thus, we collected and re-captioned 100M images with long texts. In this section, we provide details of re-captioning and explore the usage of long texts in language-image pre-training.

## 4.1 Long Text-Image Pair Dataset

**Training Dataset.**    To construct long text-image pairs for language-image pre-training, we re-captioned 100 million images with long texts. Specifically, we collected the images from CC3M [27], CC12M [27], YFCC15M [29], LAION [26], and COYO [2] dataset. Then, we instructed three multi-modality large language models (MLLMs), *i.e.*, InstructBLIP [7], LLaVA [20], and ShareGPT4V [6] to generate diverse and descriptive long texts based on the collected images. In this step, we used "Describe the image in detail." as the text prompt, following DreamLIP [41]. Finally, each image in the collected datasets is paired with four texts: a raw text from the original dataset and three re-captioned long texts. The raw texts exhibit an average length of approximately 18 tokens, whereas the re-captioned long texts consist of around 136 tokens.

**Evaluation Dataset.**    Most zero-shot evaluation tasks for language-image pre-training primarily rely on short textual input. For example, in short-text-image-retrieval tasks, the textual inputs contain fewer than 15 tokens on average, as shown in Tab. 1. Given the short texts, these tasks are not suitable for accessing the long text comprehension ability of pre-trained models. Therefore, we collected long text-image pairs from DCI [31], IIW [12], and ShareGPT4V [6] datasets to construct long-text-image retrieval evaluation tasks. Specifically, for DCI and IIW datasets, we use images with human-annotated long descriptions for retrieval. For ShareGPT4V dataset, we sample 1,000 and 10,000 data from ShareGPT4V dataset to construct ShareGPT4V-1k and ShareGPT4V-10k retrieval dataset, respectively. The ShareGPT4V-1k dataset is constructed following Long-CLIP [39]. All images in ShareGPT4V-1k and ShareGPT4V-10k are from SA-1B [15] dataset, and all long texts are generated by ShareGPT4V-Captioner [6]. As shown in Tab. 1, each image within these datasets has one paired long text, which consists of more than 8 sub-captions and 170 tokens on average. Here, a sub-caption is a complete sentence ending with a period. The long-text-image retrieval task necessitates the alignment of text features from long texts with the image features from the corresponding images, which thereby evaluates the model's ability to comprehend long texts.

## 4.2 Exploring the Influence of Text length

Long-CLIP [39] enables long text processing ability by fine-tuning CLIP model with long text-image pairs. However, the benefits and drawbacks of using long texts in pre-training are still unknown. It

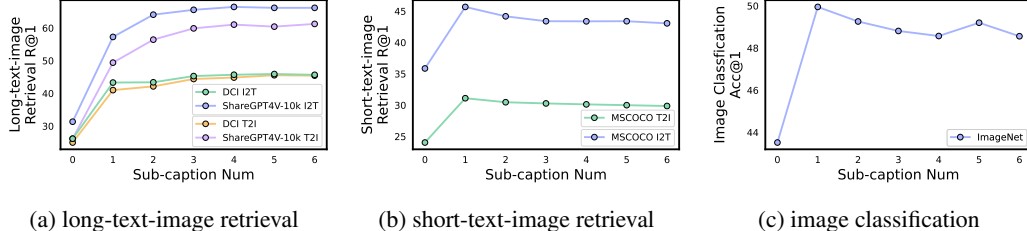

|  |  |  |
|---|---|---|
| (a) long-text-image retrieval | (b) short-text-image retrieval | (c) image classification |

Figure 2: **The influence of text length.** A significant improvement is observed across all tasks when we added one randomly sampled sub-caption from generated texts to the pre-training stage. As the number of sub-captions increases, the performance of the pre-trained model on long-text-image retrieval tasks consistently improves and becomes stable (a). However, there is a performance degradation in MSCOCO retrieval task (b) and ImageNet classification task (c).

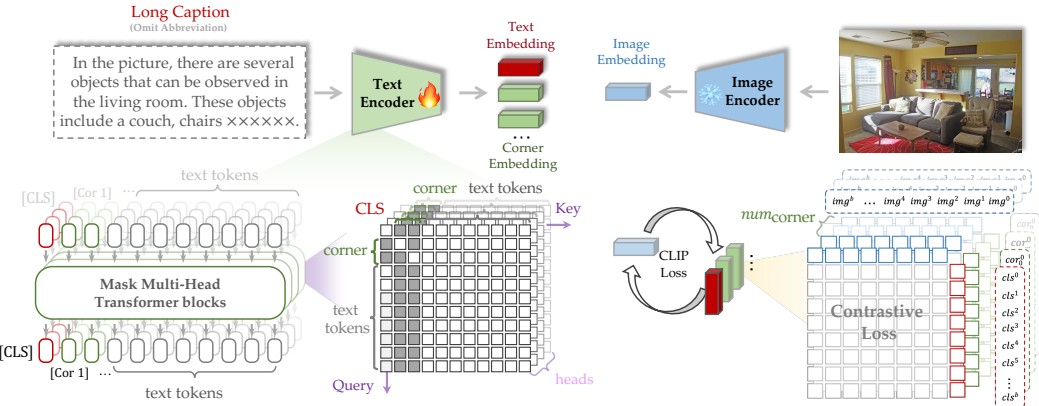

Figure 3: **Overview of** LoTLIP. We add multiple learnable corner tokens ([Cor 1], [Cor 2], $\cdots$) after [CLS] token. These corner tokens are initialized differently for aggregating diverse token features. Besides, an attention mask mechanism is used to limit the interaction between [CLS] and corner tokens to ensure the diversity of gathered features.

is also unknown how the length of the text affects the performance of the pre-trained model. To explore the influence of using texts in different lengths for pre-training, we change the length of long texts by selecting different numbers of consecutive sub-captions, as illustrated in Fig. 2. Each sub-caption in the long texts has an average of approximately 22 tokens. Sub-caption number equal to 0 indicates that the model is trained without the generated texts. When introducing one sub-caption, a noticeable gain is achieved across all tasks. Moreover, the results also indicate that pre-training with longer texts, composed of more sub-captions, enhances the performance of the pre-trained model on long-text-image retrieval tasks. It confirms that using long text-image pairs for pre-training improves the model's understanding of long texts. However, the usage of longer texts negatively impacts the model's performance on short-text-image retrieval and image classification tasks.

## 4.3 Method

In order to find a solution that well balances the long and short text understanding of the pre-trained model, we design to add extra text tokens for text encoders, termed *corner tokens*, which can aggregate diverse text features. This strategy benefits the class (*i.e.*, [CLS]) token by extracting more representative features for long and short text. Next, we will describe the details.

**Corner Tokens.** Different text encoder architectures (*e.g.*, BERT [10], T5 [25]) can be utilized in language-image pre-training. In this section, we take BERT [10] as an example of our approach. Given a long text, it is expressed as [CLS], $\cdots$ [SEP], $\cdots$ [SEP], where the first token of every text input is its class token [CLS], and all sub-captions are separated by a special token [SEP]. The omitted part in $\cdots$ denotes the tokens obtained after tokenizing words within the sub-caption. Based on the tokenized long text, we insert multiple learnable corner tokens $\mathcal{C} = \{[\text{Cor 1}], \cdots, [\text{Cor m}]\}$ after

the class token, where $m$ is the number of corner tokens. In this way, the form of the tokenized text input is $[\texttt{CLS}], [\texttt{Cor 1}] \ldots [\texttt{Cor m}], \cdots [\texttt{SEP}], \cdots [\texttt{SEP}]$. Moreover, we design an attention mask mechanism $\mathcal{A}$ for the text encoder to promote the diversity of the aggregated features. Specifically, when calculating the attention scores, the corner tokens and $[\texttt{CLS}]$ token are guided to neglect each other but attend to all other sub-caption tokens. Meanwhile, in the attention mask mechanism, each text tokens are designed to only interact with other text tokens and the $[\texttt{CLS}]$ token, to keep the interactions between local and global information. The attention mask $\mathcal{A}$ is formulated as:

$$\mathcal{A}(q, k) = \begin{cases} 0, & \text{if } (k \in \mathcal{C} \text{ or } q, k \in \mathcal{C} \cup \{[\texttt{CLS}]\}) \text{ and } q \neq k \\ 1, & \text{otherwise} \end{cases}$$

where $q$ and $k$ represent the query and key tokens in attention block, respectively. The features of the $[\texttt{CLS}]$ and corner tokens are regarded as text global feature $\mathbf{t}_g$ and corner features $\mathbf{t}_{c_1}, \mathbf{t}_{c_2}, ..., \mathbf{t}_{c_m}$, respectively.

**Optimization.** The short-text-image contrastive loss $\mathcal{L}_{\text{short}}$ is calculated in the same way as Eq. (1). Meanwhile, the long-text-image contrastive loss $\mathcal{L}_{\text{long}}$ between image global feature $v$ and $\mathbf{t}_g, \mathbf{t}_{c_1}, \mathbf{t}_{c_2}, ..., \mathbf{t}_{c_m}$ of long text is as follows:

$$\mathcal{L}_{\text{long}} = \mathcal{L}_{\text{long}}^{i2t} + \mathcal{L}_{\text{long}}^{t2i}, \tag{4}$$

$$\mathcal{L}_{\text{long}}^{i2t} = -\sum_{i=1}^{N} (\log \frac{\exp\left(\cos\langle \mathbf{v}^i, \mathbf{t}_g^i\rangle/\tau\right)}{\sum_{j=1}^{N} \exp\left(\cos\langle \mathbf{v}^i, \mathbf{t}_g^j\rangle/\tau\right)} + \sum_{k=1}^{m} \log \frac{\exp\left(\cos\langle \mathbf{v}^i, \mathbf{t}_{c_k}^i\rangle/\tau\right)}{\sum_{j=1}^{N} \exp\left(\cos\langle \mathbf{v}^i, \mathbf{t}_{c_k}^j\rangle/\tau\right)}), \tag{5}$$

$$\mathcal{L}_{\text{long}}^{t2i} = -\sum_{i=1}^{N} (\log \frac{\exp\left(\cos\langle \mathbf{t}_g^i, \mathbf{v}^i\rangle/\tau\right)}{\sum_{j=1}^{N} \exp\left(\cos\langle \mathbf{t}_g^i, \mathbf{v}^j\rangle/\tau\right)} + \sum_{k=1}^{m} \log \frac{\exp\left(\cos\langle \mathbf{t}_{c_k}^i, \mathbf{v}^i\rangle/\tau\right)}{\sum_{j=1}^{N} \exp\left(\cos\langle \mathbf{t}_{c_k}^i, \mathbf{v}^j\rangle/\tau\right)}), \tag{6}$$

The total training loss is $\mathcal{L}_{\texttt{LoTLIP}} = \mathcal{L}_{\text{long}} + \mathcal{L}_{\text{short}}$.

# 5 Experiments

## 5.1 Implementation Details and Datasets

**Pre-training Datasets.** As presented in Sec. 4.1, we collected 100M data from five publicly available image-text pair datasets and re-captioned the collected images with long texts. Based on this dataset, we construct 4 scales of pre-training data: (1) 3M, including CC3M. (2) 12M, including CC12M. (3) 30M, including CC3M, CC12M, and YFCC15M. (4) 100M, including all re-captioned data. We conduct ablation studies to validate our model on the 3M scale pre-training data. The performance of LoTLIP pre-trained with 12M and 30M scale datasets is shown in the *Supplementary Material*.

**Downstream Datasets.** To assess the ability of the pre-trained models on short-text and long-text understanding, we select 3 downstream tasks for evaluation under the zero-shot setting, including long-text-image retrieval, short-image-text retrieval, and image classification. **For long-text-image retrieval**, we present the Recall at 1 (R@1) metric of the pre-trained models on DCI, IIW, and ShareGPT4V-1k, and ShareGPT4V-10k long-text-image retrieval tasks. **For short-image-text retrieval**, we evaluate on MSCOCO [18] and Flickr30k Caption [35] and report Recall at 1/5 (R@1/5) metric for comparison. **For image classification**, we use ImageNet1k [9] for evaluation and present top-1 accuracy (Acc@1) on image classification. Following [24], we use class names incorporated with pre-defined text prompts as text inputs for zero-shot classification.

**Implementation Details.** Following LiT [37], we use a vision transformer pre-trained on ImageNet 21K as the image encoder and Bert [10] as the text encoder. The architecture of the image encoder is ViT-B/16. We report other variants of vision transformers in the *Supplementary Material*. The images are resized to $224 \times 224$. The maximum text token length is set to 128 unless specifically stated. Three consecutive sub-captions are randomly selected to form long texts as text input. We train 10 epochs on the 3M and 100M scale datasets. For the 3M dataset, the batch size is set to 2560, while that of 100M is set to 16384. The other pre-training hyperparameters are under the same setting, *e.g.* learning rate, warmup steps, and weight decay.

## 5.2 Ablation Studies

Table 2: Analyze the effectiveness of LoTLIP in language-image pre-training with long texts. The architecture of the image encoder is ViT-B/16. I2T and T2I indicate R@1 on text and image retrieval, respectively. We use 3M scale dataset for pre-training. "✓" indicates we add long texts in the training stage.

| Method | Long Texts | Long-text-image Retrieval | | | | | | Short-text-image Retrieval | | Classification |
| | | DCI | | IIW | | ShareGPT4V-10k | | MSCOCO | | ImageNet |
| | | I2T | T2I | I2T | T2I | I2T | T2I | I2T | T2I | Acc@1 |
|---|---|---|---|---|---|---|---|---|---|---|
| LiT | - | 27.14 | 24.13 | 65.20 | 58.50 | 32.73 | 27.01 | 34.20 | 24.07 | 43.76 |
| LiT | ✓ | 47.96 | 44.92 | 84.97 | 81.70 | 73.66 | 66.73 | 43.52 | 30.06 | 48.87 |
| LiT+Long-CLIP[*] | ✓ | 47.21 | 46.61 | 83.82 | 83.01 | 74.60 | 67.49 | 43.78 | 26.74 | 46.82 |
| LoTLIP | ✓ | **49.46** | **47.82** | **84.97** | **83.33** | **76.49** | **69.72** | **46.56** | **31.59** | **50.34** |

[*] We apply the primary Component matching strategy proposed by Long-CLIP [39] to LiT and train the model with the same training setting using 3M data for fair comparison.

**Exploring the Influence of Token Number Limitation.** Token number limitation is the maximum length of text input that the text encoder can process at a time. For language-image pre-training models, the token number limitation of text encoder can be arbitrarily set to any positive integer, which is typically set to 77 [24, 38, 41]. The benefits derived from using long texts in pre-training might be constrained by the 77 token limitation. To investigate this hypothesis, we incrementally raise the token number limitation from 32 to 512. The results are illustrated in Fig. 4, which indicate a limitation of 77 tokens is insufficient for a model training with long texts. On DCI and ShareGPT4V-10k retrieval tasks, the best results are observed when the token number limitation is set to 192. Meanwhile, the highest performance on the IIW retrieval task is reached

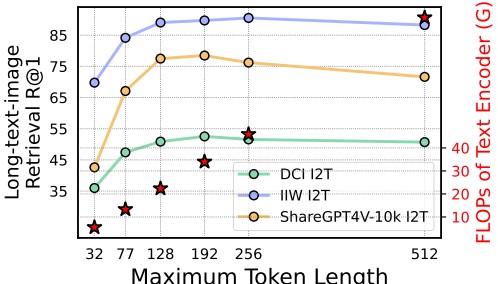

Figure 4: **Influence of token number limitation on LoTLIP.** The performance of the pre-trained model on different tasks improves when the token number limitation increases up to 192, which exceeds the commonly used 77. Meanwhile, the FLOPs of the text encoder (red stars) rapidly increase with the text token number limitation.

when the text token length limitation is 256. Since the average text length of the evaluation datasets is less than 240 tokens, as shown in Tab. 1. When the text token length limitation gets larger than 256, these datasets can't well prove the long-text understanding ability of the model. In Fig. 4, we also illustrate the FLOPs of the text encoder, which increases with the text token number limitation. To balance the training efficiency and performance, we set the token number limitation to 128 for the text encoder of LoTLIP.

**Compare LoTLIP with Other Methods in Pre-training with Long Texts.** LoTLIP aims to enhance the long-text and short-text comprehension ability of language-image pre-training. To demonstrate the effectiveness of LoTLIP, we compare it with other approaches that train with long text-image pairs. The first approach is directly using long texts in the training stage of LiT. The second approach is based on another contrastive learning method related to long texts, namely Long-CLIP [39]. Concretely, we incorporate the primary component matching strategy and losses proposed by Long-CLIP to LiT (LiT+Long-CLIP). For a fair comparison, all models are trained with 3M scale dataset. As shown in Tab. 2, directly using long texts in pre-training procedural significantly improves LiT over all tasks. Moreover, LiT+Long-CLIP exceeds LiT on the long-text-image retrieval tasks when both models incorporate long texts during pre-training. But on short-text comprehension tasks, *i.e.*, short-text-image retrieval and image classification, the performance of LiT+Long-CLIP is inferior to that of LiT. Instead, LoTLIP improves LiT and LiT+Long-CLIP on both long and short text comprehension tasks. Concretely, LoTLIP surpasses LiT and LiT+Long-CLIP by 1.97% and 1.51% on average over three long-text-image retrieval tasks. Besides, LoTLIP improves LiT by 2.29% and 1.47% on MSCOCO retrieval and ImageNet classification, respectively. The results demonstrate

Table 3: Analyze the influence of the number of corner tokens and the attention mask mechanism. We use 3M scale dataset for training. The architecture of the image encoder is ViT-B/16.

| #Corner Token | Attention Mask | Long-text-image Retrieval | | | | | | Short-text-image Retrieval | | Classification |
| | | DCI | | IIW | | ShareGPT4V-10k | | MSCOCO | | ImageNet |
| | | I2T | T2I | I2T | T2I | I2T | T2I | I2T | T2I | Acc |
|---|---|---|---|---|---|---|---|---|---|---|
| 0 | - | 47.96 | 44.92 | 84.97 | 81.70 | 73.66 | 66.73 | 43.52 | 30.06 | 48.87 |
| 1 | ✓ | 49.57 | 46.55 | 84.97 | 82.68 | 74.91 | 68.41 | 45.68 | 31.51 | 49.62 |
| 2 | ✓ | 49.46 | 47.82 | 84.97 | 83.33 | 76.49 | 69.72 | 46.56 | 31.59 | 50.34 |
| 3 | ✓ | **49.58** | 47.70 | **87.09** | **84.31** | **76.51** | **70.20** | 46.48 | 31.60 | 50.36 |
| 4 | ✓ | **49.58** | 48.30 | 86.76 | 84.15 | 76.25 | 70.14 | **47.70** | **31.88** | **50.59** |
| 2 | - | 48.61 | 47.17 | 86.11 | 81.86 | 76.14 | 69.31 | 47.70 | 31.34 | 49.88 |
| 2 | ✓ | 49.46 | 47.82 | 84.97 | 83.33 | 76.49 | 69.72 | 46.56 | 31.59 | 50.34 |

Table 4: Zero-shot evaluation of different models on long-text-image retrieval tasks. I2T and T2I indicate R@1 on text and image retrieval, respectively.

| Data Scale | | Model | DCI | | IIW | | ShareGPT4V-1k | | ShareGPT4V-10k | | Avg. |
| Short | Long | | I2T | T2I | I2T | T2I | I2T | T2I | I2T | T2I | |
|---|---|---|---|---|---|---|---|---|---|---|---|
| 3M | - | FILIP [34] | 10.85 | 11.36 | 31.54 | 29.08 | 26.50 | 26.80 | 8.94 | 8.64 | 19.21 |
| 3M | - | LaCLIP [11] | 14.84 | 14.71 | 41.01 | 38.89 | 40.90 | 37.10 | 15.81 | 14.84 | 27.26 |
| 3M | - | SigLIP [38] | 11.66 | 13.11 | 29.25 | 29.58 | 27.30 | 25.10 | 9.92 | 9.30 | 19.40 |
| 3M | - | LiT [37] | 27.14 | 24.13 | 65.20 | 58.50 | 63.60 | 56.80 | 32.73 | 27.01 | 44.38 |
| 3M | | LoTLIP | **49.46** | **47.82** | **84.97** | **83.33** | **93.20** | **90.00** | **76.49** | **69.72** | **74.37** |
| 400M | - | CLIP [24] | 45.45 | 43.01 | 88.24 | 87.58 | 84.50 | 79.80 | 60.22 | 56.16 | 68.12 |
| 100M | - | LiT [37] | 41.78 | 40.90 | 88.07 | 82.68 | 86.00 | 80.00 | 61.41 | 50.69 | 66.44 |
| 700M | - | ALIGN [14] | 56.54 | 57.41 | 92.65 | 90.68 | 86.30 | 85.30 | 65.13 | 62.73 | 74.59 |
| 12B | - | SigLIP [38] | 57.78 | 56.22 | 91.99 | 91.01 | 85.80 | 83.40 | 83.40 | 63.08 | 76.59 |
| 400M | 1M | Long-CLIP* [39] | 51.68 | 57.28 | 89.61 | **93.20** | 94.70 | 93.40 | 79.24 | 77.06 | 79.52 |
| 100M | | LoTLIP | **62.10** | **61.06** | **93.95** | 92.48 | **96.50** | **95.50** | **86.84** | **81.40** | **83.72** |

* Long-CLIP fine-tunes pre-trained CLIP model with ShareGPT4V dataset except ShareGPT4V-1k.

that LoTLIP benefits from long texts and corner tokens, thereby exhibiting a better understanding of the long and short texts.

**Implementation of Corner Tokens.** In this part, we study the influence of the attention mask mechanism and the number of corner tokens on different downstream tasks as shown in Tab. 3. LoTLIP pre-trained without the pre-defined attention mask encounters performance degradation on most tasks compared to when using the pre-defined attention mask. It indicates that direct interaction between the CLS token and corner tokens limits their ability to aggregate diverse textual features. On short-text-image retrieval and image classification tasks, the performance of LoTLIP improves when introducing more corner tokens. It proves that the corner tokens help with enhancing the short text understanding ability of LoTLIP. When the number of corner tokens is set to more than two, the performance improvement across all tasks is relatively small. Thus, we use two corner tokens in LoTLIP.

## 5.3 Main Results

We compare LoTLIP with the state-of-the-art methods on downstream tasks involving long texts and short texts. The experimental results are shown in Tab. 4 and Tab. 5, respectively. On 3M data scale, LoTLIP significantly improves the state-of-the-art methods on all tasks. Specifically, LoTLIP

Table 5: Zero-shot evaluation of different models on short-text-image retrieval and classification tasks.

| Data Scale | | Model | IN | MSCOCO I2T | | MSCOCO T2I | | Flickr30k I2T | | Flickr30k T2I | |
| Short | Long | | Acc. | R@1 | R@5 | R@1 | R@5 | R@1 | R@5 | R@1 | R@5 |
|---|---|---|---|---|---|---|---|---|---|---|---|
| 3M | - | FILIP [34] | 18.69 | 14.98 | 34.88 | 11.86 | 28.98 | 30.40 | 56.80 | 20.76 | 44.20 |
| 3M | - | LaCLIP [11] | 21.50 | 18.94 | 40.40 | 12.42 | 31.04 | 37.00 | 63.90 | 29.32 | 56.04 |
| 3M | - | SigLIP [38] | 21.28 | 16.30 | 37.36 | 13.22 | 31.65 | 31.00 | 61.10 | 23.76 | 48.64 |
| 3M | - | LiT [37] | 43.76 | 34.20 | 61.52 | 24.07 | 48.37 | 61.30 | 89.50 | 48.06 | 75.36 |
| 3M | | LoTLIP | **50.34** | **46.56** | **72.02** | **31.59** | **57.65** | **75.20** | **94.20** | **58.20** | **83.58** |
| 400M | - | CLIP [24] | 68.34 | 51.68 | 76.72 | 32.70 | 57.76 | 82.20 | 96.60 | 62.14 | 85.72 |
| 100M | - | LiT [37] | 73.70 | 51.92 | 75.72 | 32.74 | 57.84 | 80.00 | 95.90 | 60.86 | 84.62 |
| 700M | - | ALIGN [14] | 65.89 | 60.42 | 82.50 | 42.36 | 67.38 | 88.90 | 98.00 | 74.12 | **92.40** |
| 12B | - | SigLIP [38] | **76.04** | **65.46** | **86.22** | **47.14** | **72.10** | **89.10** | **98.00** | **74.66** | 92.30 |
| 400M | 1M | Long-CLIP [39] | 67.10 | 57.28 | 80.78 | 40.34 | 65.92 | 85.90 | 98.50 | 70.66 | 90.60 |
| 100M | | LoTLIP | 72.16 | 59.66 | 81.50 | 38.06 | 63.81 | 86.90 | 97.80 | 65.22 | 87.98 |

improves the second-best method LiT by 29.99% on average over four long-text-image retrieval tasks. Moreover, LoTLIP improves the second competitor LiT by 6.58% and 10.98% on image classification task and short-text-image retrieval tasks, respectively. It proves that LoTLIP significantly enhances the language-image model for understanding short and long captions by involving long captions in pre-training and incorporating corner tokens in text inputs. It is worth noting that LoTLIP trained with 100M data exceeds all state-of-the-art methods on long-text-image retrieval tasks, even though these methods are pre-trained on a larger scale of data. Concretely, LoTLIP (trained on 100M data) exceeds SigLIP (trained with 12B) by an average of 7.13% over four long-text-image retrieval tasks. Moreover, compared to Long-CLIP, which uses long texts in CLIP for fine-tuning, LoTLIP improves averaged performance by 4.2% on long-text-image retrieval tasks.

## 6  Conclusion

In this work, we empirically confirm that a key issue arises because training images are typically paired with short captions, which can cause certain tokens to be overshadowed by more salient ones. To address this issue, our initial strategy involved relabeling the data with long captions. However, directly learning from these long captions might lead to degraded performance in tasks requiring an understanding of short text, such as image classification. Subsequently, by incorporating corner tokens to aggregate diverse textual information, we are able to help the model regain its original proficiency in understanding short texts while significantly enhancing its capability to comprehend long texts. We also investigated whether the model could continue to benefit from longer captions and observed a clear trade-off between performance and efficiency. Finally, we validated the effectiveness of our approach using a self-constructed large-scale dataset, which consists of 100 million long-caption-oriented text-image pairs. In the task of long-text-image retrieval, our method outperforms the second-best competitor, Long-CLIP, with an improvement of 4.2%.

## 7  Acknowledgments

This work was supported by National Natural Science Foundation of China (NSFC) under Grants 62225207, and Zhejiang Provincial Natural Science Foundation of China under Grants LD24F020011.

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

# 8 Appendix

## 8.1 Data Statistics

As shown in Tab. 6, we report the data statistics of our dataset and other text-image paired datasets. Compared to the other text-image paired datasets, the re-captioned texts of LoTLIP are significantly longer, averaging 136 tokens *v.s.* 18 tokens. To the best of our knowledge, LoTLIP dataset is the largest dataset consisting of long texts for multi-modal learning. We are continuing to expand the size of LoTLIP by integrating additional MLLMs, as well as gathering more publicly available datasets for long text generation.

Table 6: Data statistic of LoTLIP dataset and other text-image paired dataset. Our dataset is the largest dataset consisting of long texts for multi-modal learning.

| Dataset | #Images | #Texts | #Sub-captions per Text | #Tokens per Text |
|---|---|---|---|---|
| Short-text-image Pairs Dataset | | | | |
| CC3M [27] | 3,018,175 | 3,018,175 | 1.01 | 11.29 |
| CC12M [27] | 10,445,969 | 10,445,969 | 1.00 | 17.48 |
| YFCC15M [29] | 14,772,456 | 14,772,456 | 1.23 | 13.61 |
| LAION47M [26] | 49,677,119 | 49,677,119 | 1.28 | 18.99 |
| COYO24M [2] | 24,658,004 | 24,658,004 | 1.21 | 17.07 |
| Long-text-image Pairs Dataset | | | | |
| LoTLIP | 102,571,723 | 307,715,169 | 6.16 | 136.14 |

## 8.2 More Experimental Analysis

### 8.2.1 Influence of Sub-caption Number on `LoTLIP` and LiT

During the pre-training stage, we randomly select multiple consecutive sub-captions from long texts in LoTLIP as long text inputs. In Fig. 5, we show the influence of the number of sub-captions when the length of text input is limited to 128. With the number of sub-captions increasing, LiT and LoTLIP reaches better performance in both text retrieval and image retrieval on the ShareGPT4V-10k and DCI datasets. There are small improvements when the number of sub-captions gets larger than 3. Moreover, both methods exhibit performance degradation on text-image retrieval and image classification tasks as the number of sub-captions increases. But the performance drop of `LoTLIP` is even slower. It further indicates that `LoTLIP` can enhance the understanding of long texts while retaining the short-text understanding ability.

### 8.2.2 Compare Corner Tokens with Register Tokens

In `LoTLIP`, we employ learnable tokens, namely corner tokens, to help the model regain its original performance in understanding short texts, while significantly enhancing its capability to comprehend long texts. It is mentioned that registered token [8] also adds learnable tokens in the encoder, which are used to get rid of artifacts in image feature maps. In order to fairly compare corner tokens with register tokens, we implement the same number of register tokens on the text encoder and train the model with the same scale of dataset as `LoTLIP`. As shown in Tab. 7, the results demonstrate that corner tokens improve the performance of the model for long-text and short-text understanding tasks, which is better than register tokens.

### 8.2.3 Contrastive Learning without Pre-trained Weights

In Tab. 8, we present the performance of `LoTLIP` without loading pre-trained weights for text and image encoder on long and short-text related tasks. The results show that directly involving long texts in the pre-training stage can also enhance the performance of the language-image model with randomly initialized weights over all tasks. Besides, `LoTLIP` further improves CLIP on long and short text-related tasks, when they are both pre-trained with long texts. In our methodology, we opt

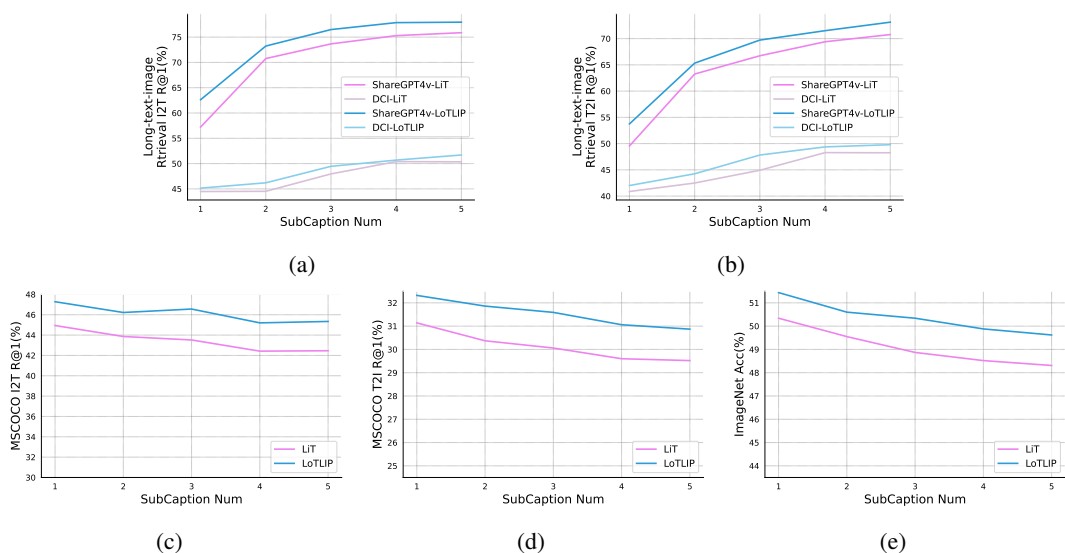

Figure 5: Influence of the number of sub-captions used in the pre-training stages. Both LiT and LoTLIP are trained with long texts. The performance on ShareGPT4V and DCI retrieval are shown in (a)(b). (c)(d) represent the performance on MSCOCO retrieval. (e) shows the performance of image classification on ImageNet.

Table 7: Compare corner tokens with register tokens. The models are trained with 3M scale dataset.

| Learnable Token Type | Long-text-image Retrieval | | | | | | Short-text-image Retrieval | | Classification |
| | DCI | | IIW | | ShareGPT4V-10k | | MSCOCO | | ImageNet |
| | I2T | T2I | I2T | T2I | I2T | T2I | I2T | T2I | Acc. |
|---|---|---|---|---|---|---|---|---|---|
| - | 47.96 | 44.92 | 84.97 | 81.70 | 73.66 | 66.73 | 43.52 | 30.06 | 48.87 |
| Register | 45.62 | 44.65 | 83.99 | 81.37 | 74.42 | 68.78 | 44.18 | 30.34 | 48.78 |
| Corner | 49.46 | 47.82 | 84.97 | 83.33 | 76.49 | 69.72 | 46.56 | 31.59 | 50.34 |

to load pre-trained weights and fix the image encoder during training, following LiT. This is because LiT is a strong baseline and such a training setting significantly conserves computational resources.

Table 8: Analyze the effectiveness of LoTLIP without loading pre-trained weights for image and text encoder. The architecture of image encoder is ViT-B/16. We use 3M scale dataset for pre-training. "✓" indicates we use long texts in the training stage.

| Method | Long Texts | Long-text-image Retrieval | | | | | | Short-text-image Retrieval | | Classification |
| | | DCI | | IIW | | ShareGPT4V-10k | | MSCOCO | | ImageNet |
| | | I2T | T2I | I2T | T2I | I2T | T2I | I2T | T2I | Acc. |
|---|---|---|---|---|---|---|---|---|---|---|
| CLIP | - | 11.67 | 11.01 | 33.17 | 31.37 | 10.69 | 8.77 | 14.74 | 10.93 | 16.54 |
| CLIP | ✓ | 42.92 | 42.23 | 77.29 | 75.98 | 66.78 | 64.70 | 32.14 | 22.33 | 23.28 |
| LoTLIP | ✓ | **43.91** | **43.83** | **78.76** | **75.98** | **70.20** | **67.91** | **34.40** | **24.11** | **24.65** |

### 8.2.4 Utilizing Long Captions from Different MLLMs

We use three MLLMs, *i.e.*, InstructBLIP [7], LLaVA [20], and ShareGPT4V [6], to generate diverse long captions for 100M images. In this part, we conduct experiments to analyze the influence of using long texts generated by different MLLMs in pre-training. As shown in Tab. 9, LoTLIP trained with all generated long texts reach the best performance on average over all tasks. It means that diverse

long texts help language-image pre-trained models to better comprehend texts to align with images. Moreover, `LoTLIP` trained with long captions generated by ShareGPT4V reaches higher scores on long-text-image retrieval tasks than other MLLMs. This indicates that the long captions generated by ShareGPT4V are of higher quality, which can enhance the long-text comprehension ability of `LoTLIP`.

Table 9: Utilizing long captions generated by different MLLMs in the training statge.

| Method | MLLM | Long-text-image Retrieval | | | | | | Short-text-image Retrieval | | Classification |
| | | DCI | | IIW | | ShareGPT4V-10k | | MSCOCO | | ImageNet |
| | | I2T | T2I | I2T | T2I | I2T | T2I | I2T | T2I | Acc. |
|---|---|---|---|---|---|---|---|---|---|---|
| `LoTLIP` | InstructBLIP [7] | 39.94 | 38.54 | 78.43 | 75.33 | 57.90 | 55.17 | 44.60 | 30.94 | 50.32 |
| | LLaVA [20] | 41.67 | 40.37 | 78.26 | 72.22 | 54.50 | 50.49 | 44.74 | 31.10 | 48.10 |
| | ShareGPT4V [6] | 47.21 | 46.71 | 85.78 | 83.17 | 77.93 | 72.30 | 44.98 | 30.96 | 49.34 |
| | All | 49.46 | 47.82 | 84.97 | 83.33 | 76.49 | 69.72 | 46.56 | 31.59 | 50.34 |

### 8.2.5 Pre-training with different scale of dataset

We compare `LoTLIP` and LiT on different scales of the pre-training dataset, *i.e.*, 3M, 12M, 30M, and 100M on long-text-image retrieval, text-image retrieval, and image classification. The results on long-text and short-text centered tasks are shown in Tab. 10 and Tab. 11, respectively. `LoTLIP` significantly improves LiT on all tasks in both ViT-B/32 and ViT-B/16 under different pre-training settings. It proves that involving long captions in the pre-training stage can help the language-image model to better deal with both long and short-comprehension tasks.

### 8.3 Visualization

We visualize the attention map of LiT, LiT trained with long texts, Long-CLIP (our implementation), and `LoTLIP` in Fig. 6. Given the long caption in the first column, the attention visualization map of LiT can only activate regions corresponding to partial objects mentioned in the long caption, *e.g.*, flat rock. This situation is alleviated after incorporating long captions in LiT pre-training. Benefiting from corner tokens, the highlighted image regions of `LoTLIP` are well aligned with the given long caption.

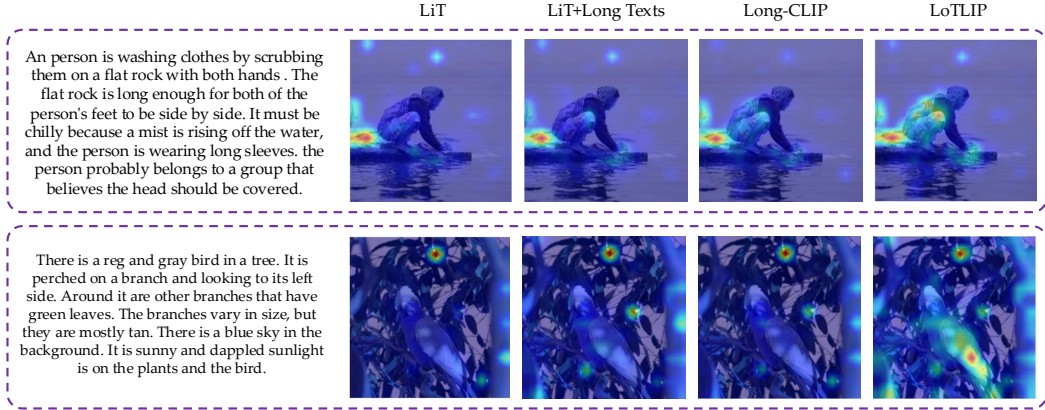

Figure 6: Visualize the attention map of LiT, LiT trained with long texts (LiT+Long Texts), Long-CLIP, and `LoTLIP`. Here, both Long-CLIP (our implementation) and `LoTLIP` are trained with long texts. Benefiting from long texts and corner tokens, the highlighted image regions of `LoTLIP` are better aligned with the given long caption compared to other methods.

### 8.4 Limitation

We re-wright the captions of 100M images using three popular open-sourced multi-modal large language models (*i.e.*InstructBLIP [7], LLaVA [20] and ShareGPT4V [6]), but we observed hallucination elements in the synthesized long captions. The hallucinations in captions, which do not correspond with the image information, may restrict the full potential of long captions in enhancing the understanding of lengthy texts by language-image pre-trained models.

### 8.5 Experiment Hardware

To obtain our LoTLIP trained with 100M scale dataset, we apply A100 GPU with 80G memory for training, which costs about 133 GPU days. For models trained on datasets of other scales (*i.e.*3M, 12M, 30M), the training duration decreases linearly with the amount of training data.

### 8.6 Social Impact

The 100M images used for pre-training are publicly accessible, and the re-annotated long captions are synthesized using public multi-modal large language models on these public datasets, posing no ethical risk in the data source. Our LoTLIP is incapable of generating images and text, thus there is no need for concern regarding the negative social impact resulting from fake, violent, pornographic, or discriminatory content. Moreover, LoTLIP has the potential for positive social impact, considering its excellent image-text retrieval performance, it may serve as a valuable asset in image retrieval libraries in the future.

### 8.7 Ethic Statement

We have conducted a thorough review to ensure that there has been no violation of the NeurIPS Code of Ethics in this paper.

Table 10: Zero-shot evaluation of different models on long-text-image retrieval.

| Data Scale | Model | DCI | | IIW | | ShareGPT4V-1k | | ShareGPT4V-10k | | Avg. |
|---|---|---|---|---|---|---|---|---|---|---|
| | | I2T | T2I | I2T | T2I | I2T | T2I | I2T | T2I | |
| *Model Architecture: ViT-B/32* | | | | | | | | | | |
| 3M | LiT | 24.11 | 20.56 | 59.48 | 53.92 | 57.80 | 51.00 | 28.19 | 22.73 | 39.72 |
| | LoTLIP | 44.46 | 41.78 | 83.99 | 79.90 | 91.40 | 87.10 | 71.17 | 63.85 | 70.45 |
| 12M | LiT | 38.73 | 33.91 | 78.59 | 77.12 | 73.00 | 69.20 | 43.69 | 38.27 | 56.56 |
| | LoTLIP | 47.35 | 45.86 | 89.87 | 87.58 | 93.80 | 91.90 | 79.01 | 73.00 | 76.04 |
| 30M | LiT | 39.41 | 33.71 | 85.13 | 76.63 | 79.40 | 67.70 | 52.09 | 36.47 | 58.82 |
| | LoTLIP | 51.12 | 49.25 | 89.87 | 89.05 | 94.40 | 92.80 | 82.66 | 76.93 | 78.26 |
| 100M | LiT | 42.54 | 39.12 | 86.27 | 83.17 | 82.10 | 76.90 | 57.69 | 47.99 | 64.47 |
| | LoTLIP | 58.64 | 55.20 | 92.81 | 91.67 | 95.10 | 93.10 | 82.81 | 77.46 | 80.84 |
| *Model Architecture: ViT-B/16* | | | | | | | | | | |
| 400M+1M | Long-CLIP | 51.68 | 57.28 | 89.61 | 93.20 | 94.70 | 93.40 | 79.24 | 77.06 | 79.52 |
| 3M | LiT | 27.14 | 24.13 | 65.20 | 58.50 | 63.60 | 56.80 | 32.73 | 27.01 | 44.38 |
| | LoTLIP | 49.46 | 47.82 | 84.97 | 83.33 | 93.20 | 90.00 | 76.49 | 69.72 | 74.37 |
| 12M | LiT | 44.04 | 39.86 | 83.01 | 80.88 | 79.60 | 74.80 | 52.93 | 45.79 | 62.61 |
| | LoTLIP | 52.24 | 50.72 | 92.16 | 89.87 | 95.80 | 93.80 | 83.78 | 77.51 | 79.48 |
| 30M | LiT | 38.68 | 34.81 | 85.29 | 79.41 | 81.40 | 68.50 | 54.63 | 39.27 | 60.24 |
| | LoTLIP | 56.04 | 55.90 | 93.79 | 91.50 | 96.90 | 94.10 | 86.72 | 81.57 | 82.06 |
| 100M | LiT | 41.78 | 40.90 | 88.07 | 82.68 | 86.00 | 80.00 | 61.41 | 50.69 | 66.44 |
| | LoTLIP | 62.10 | 61.06 | 93.95 | 92.48 | 96.50 | 95.50 | 86.84 | 81.40 | 83.72 |

Table 11: Zero-shot evaluation of different models on image-text retrieval and classification tasks.

| Data Scale | Model | IN Acc. | MSCOCO I2T R@1 | MSCOCO I2T R@5 | MSCOCO T2I R@1 | MSCOCO T2I R@5 | Flickr30k I2T R@1 | Flickr30k I2T R@5 | Flickr30k T2I R@1 | Flickr30k T2I R@5 |
|---|---|---|---|---|---|---|---|---|---|---|
| colspan | | *Model Architecture: ViT-B/32* | | | | | | | | |
| 3M | LiT | 38.83 | 30.94 | 57.02 | 21.25 | 44.60 | 44.60 | 83.80 | 41.66 | 71.70 |
| | LoTLIP | 45.50 | 41.98 | 68.44 | 28.00 | 53.36 | 72.00 | 91.10 | 52.76 | 79.16 |
| 12M | LiT | 59.84 | 40.70 | 66.52 | 26.48 | 51.47 | 66.90 | 89.00 | 49.52 | 76.58 |
| | LoTLIP | 61.07 | 50.84 | 76.20 | 33.19 | 58.89 | 77.30 | 94.60 | 57.46 | 83.08 |
| 30M | LiT | 64.95 | 42.48 | 68.36 | 25.70 | 50.50 | 67.80 | 89.80 | 47.64 | 76.14 |
| | LoTLIP | 65.41 | 52.52 | 76.72 | 32.51 | 57.90 | 78.40 | 95.00 | 56.16 | 81.14 |
| 100M | LiT | 68.26 | 47.98 | 72.80 | 29.69 | 55.24 | 75.60 | 93.90 | 54.52 | 81.34 |
| | LoTLIP | 67.20 | 55.62 | 79.30 | 34.98 | 60.80 | 82.60 | 95.30 | 59.58 | 84.56 |
| colspan | | *Model Architecture: ViT-B/16* | | | | | | | | |
| 400M+1M | Long-CLIP | 67.10 | 57.28 | 80.78 | 40.34 | 65.92 | 85.90 | 98.50 | 70.66 | 90.60 |
| 3M | LiT | 43.76 | 34.20 | 61.52 | 24.07 | 48.37 | 61.30 | 89.50 | 48.06 | 75.36 |
| | LoTLIP | 50.34 | 46.56 | 72.02 | 31.59 | 57.65 | 75.20 | 94.20 | 58.20 | 83.58 |
| 12M | LiT | 65.97 | 44.78 | 70.46 | 29.57 | 54.58 | 72.90 | 92.90 | 54.18 | 81.18 |
| | LoTLIP | 66.70 | 55.18 | 78.54 | 36.22 | 62.05 | 82.60 | 96.30 | 62.74 | 86.78 |
| 30M | LiT | 69.91 | 45.46 | 70.48 | 27.66 | 51.96 | 73.40 | 93.10 | 53.16 | 78.94 |
| | LoTLIP | 70.60 | 55.58 | 79.68 | 34.34 | 59.85 | 81.80 | 95.70 | 60.52 | 84.22 |
| 100M | LiT | 73.70 | 51.92 | 75.72 | 32.74 | 57.84 | 80.00 | 95.90 | 60.86 | 84.62 |
| | LoTLIP | 72.16 | 59.66 | 81.50 | 38.06 | 63.81 | 86.90 | 97.80 | 65.22 | 87.98 |

