# OpenReview forum: "LoTLIP: Improving Language-Image Pre-training for Long Text Understanding"
_NeurIPS.cc/2024/Conference — NeurIPS 2024 poster_

### Official Review · Reviewer_NJUx · 2024-07-04

**Soundness:** 2
**Presentation:** 1
**Contribution:** 2
**Rating:** 5
**Confidence:** 4

**Summary:**

This paper introduces LotCLIP, which enhances CLIP’s capability to understand long texts. It highlights that merely increasing the length of texts (i.e., context length) is not beneficial as it adversely impacts the understanding of short texts (i.e., image classification). To mitigate this trade-off, additional learnable corner tokens are integrated into the text encoder transformer. Moreover, the attention mask is modified so that interactions between corner tokens are restricted, ensuring diversity in the output feature embeddings. The image encoder and text encoder are initialized from pre-trained models and further trained in a LiT manner. LotCLIP is trained on a self-constructed image-text dataset of 100M scale long texts using MLLMs. LotCLIP is evaluated on both long-text retrieval tasks and short-text benchmarks such as image-text retrieval and classification.

**Strengths:**

- This paper addresses the underexplored yet important problem of enhancing CLIP's ability to understand long texts. It introduces a simple modification to the CLIP training framework.

- The 100M scale long caption data will be valuable for training VLMs with an understanding of long contexts.

- Strong empirical results compared to other baselines such as LiT , CLIP, and SigLIP.

**Weaknesses:**

- [W1] Contributions are unclear. From an architectural perspective, there seems to be no major difference from [1, 2], which introduce additional learnable tokens in the encoders. Is there any special reason or evidence that such a technique is especially beneficial for training with long captions? It is likely to be a general technique that is also helpful for training with short captions. Such learnable tokens, named corner tokens in the paper, are only prepended to the text tokens after the [CLS] token. Corner tokens in different positions, such as after the text tokens, can be further ablated. Would corner tokens also be helpful in vision transformers? More comprehensive analysis around the corner tokens are necessary.

- [W2] Connected to the [W1], the contributions of the data are unevaluable. It only mentions that some image-text pairs from various sources (e.g., CC3M/12M, YFCC15M, LAION, and COYO) are re-captioned using MLLMs such as InstructBLIP, LLaMA, and ShareGPT-4V. No other clear details are provided. Any statistics about the training data are missing. Any details on the MLLMs such as specific architecture and instruction information are completely omitted. For the LAION and COYO datasets, how is a subset selected from the entire scale? Most importantly, no verification step for the extracted long captions is provided. Due to the typical hallucinations of MLLMs, it is unclear how well the obtained long captions align with the original images or original short captions, further complicating the training of VLMs.

- [W3] It is unclear whether the comparison is fair. With the LiT training mechanism, LotCLIP benefits from an ImageNet-pretrained ViT backbone, which shows strong evaluation results on ImageNet compared to other pretrained backbones. It is suggested that other visual backbones pretrained via CLIP or through unsupervised methods such as DINO be tested with LiT training.

- [W4] Training and some evaluation benchmarks seem to overlap, leading to high performance results. For example, ShareGPT has LAION and Conceptual Captions images, which shares the training data. This creates a significant gap in evaluation results compared to DCI, and even between models from the default CLIP and the proposed LotCLIP in ShartGPT evaluation.

- [W5] It is not directly comparable to LongCLIP. Starting from the same pretrained CLIP model, how does LotCLIP perform when fine-tuned on a 1M-scale dataset similar to that used for LongCLIP?


In summary, in the current version, there is no clear evidence of technical contributions and the experimental settings are unclear.


---

References

[1] Darcet et al., Vision Transformers Need Registers, in ICLR 2024.

[2] Lavoie et al., Modeling Caption Diversity in Contrastive Vision-Language Pretraining, in arXiv preprint 2024.

**Questions:**

- The reasoning behind the naming of the constructed data and model is unclear. Why are they named Dora and LotCLIP?

- In the introduction section, the concept of the corner token first appears, but without supporting explanations, which creates confusion.

- No training details for the baseline methods are provided.

**Limitations:**

- Some limitations on the long-caption data (e.g., hallucinations) are mentioned.

---

> ### Author Rebuttal · Authors · 2024-08-07
>
> ``Q1: Contributions are unclear.``
>
> Sorry for confusion. We reaffirm our core contribution:
> - **We are the first to explore how to improve understanding long texts in contrastive language-image pre-training, and also firstly design learnable text [CLS] tokens (corner tokens) for this purpose.**
>
> Meanwhile, we heartily believe that it is necessary to study how to design a text tencoder that is visually aligned and has the ability to understand long texts.
>
> ``Q2: From an architectural perspective, there seems to be no major difference from [1, 2].``
>
> We are beg to differ with you on this matter. The core difference is not learnable tokens, but **text [CLS] tokens and our designed attention mask mechanism.** [1] and [2] both design learnable tokens in image encoder, which give few help to long-text understanding (refer to Reviewer NJUx, Q4). Moreover, directly **introducing register tokens [1] to text encoder** also does not improve the performance **(-0.02 performance degradation)**, because register tokens is used to get rid of artifacts in images (This problem not exists in text). Our coner token can enhance its capability of long text understanding **(+2.05 performance improvment)**
>
> As a concurrent work with our LotCLIP, we look forward to whether using [2] into text encoder can enhance CLIP's ability to understand long texts, but unfortunately its official code is currently inaccessible.
>
> ``Q3: Is there any special reason or evidence that such a technique is especially beneficial for training with long captions?``
>
> This is really an interesting aspect. **Coner token is a general technique but significantly enhances the understanding of long text.** Moreover, our method shows a more significant performance improvement when applied to long captions (+2.05 performance improvment) compared to shorter ones (+0.18 performance improvment).
>
> `` Q4: Corner tokens in different position``
>
> Thanks for you valuable suggestion. **Corner tokens around [CLS] token may bring better long-text understanding capacity.**  After Text = 67.82%,  Before [CLS] =68.49% **(+0.67)**,  After [CLS] = 68.63% **(+0.81)**. In BERT architecture, the [CLS] token is always the first token of text input. Thus, around [CLS] token may have better performance.
>
> `` Q5: Corner tokens in vision transformers``
>
> Thanks for your valuable suggestion. We implement the corner tokens in the image encoder and find that it provides less improvement (0.99% performance improvement) to long-text understanding compared to utilize corner tokens in the text encoder (2.01% performance improvement).
>
> `` Q6: Statistics about the training data``
>
> As shown in the table 1 of global pdf, we report some statistics of our dataset and compare it with other long-text datasets.
>
> `` Q7: Details on the MLLMs such as specific architecture and instruction information``
>
> We  provide the hyper-parameter settings of the used MLLMs as shown in the table 2 of global pdf.
>
> `` Q8: For the LAION and COYO datasets, how is a subset selected?``
>
> Following Stable Diffusion, we filter Laion and COYO to images with aesthetic_v2>5.0 and resolution ≥512 to about 70M preserved pairs.
>
> `` Q9: Verification step for the extracted long captions.``
>
> Thanks for your suggestion. We verify the quality of long texts of three MLLM for  IIW dataset, and compare with human annotated long texts. We utilize GPT4V for assessment the alignment between image and text, following Q-Bench [3] (IIW =4.51, InstructBLIP-Vicuna7B=2.80, LLaVA-v1.5-13B=3.45, ShareGPT4V-13B=3.48). They have good score and low hallucination. Training with long texts from multiple MLLMs may mitigate the inherent biases of one MLLM.
>
> [3] Q-Bench: A Benchmark for General-Purpose Foundation Models on Low-level Vision
>
> `` Q10: It is unclear whether the comparison is fair. More visual backbones with LiT training``
>
> We compared various methods as fairly as possible. **Our long-caption dataset and designed corner token can consistently improve various artitectures.** We use unsupervised ViT (DINO) and self-supervised ViT (MoCo-v3) as image encoder and counduct training with 3M scale dataset following LiT.
>
> |Method| Pretrained Visual Backbone  |ImageNet1k Cls | Long text-image Retrival Avg.|
> |  :----: |  :----: | :----: | :----: |
> CLIP| - | 16.54 |17.78|
> LotCLIP|-|24.65 **(+8.11)**|63.43 **(+45.65)**|
> LiT | MoCo-v3 |  35.52|32.48 |
> LotCLIP | MoCo-v3 | 41.27 **(+5.75)**|59.99 **(+27.51)**|
> LiT | DINO |37.16| 34.28 |
> LotCLIP | DINO | 42.43 **(+5.27)**| 59.24 **(+24.96)**|
>
> `` Q11: Overlap on training and evaluation dataset``
>
> **There is no overlap in the images** used for training and the ShartGPT4V evaluation, as all the images in **our used ShartGPT4V evaluation are sourced from the SAM dataset**.
>
> `` Q12: Directly comparable to LongCLIP``
>
> Thanks for your valuable suggestion. **Directly comparable to LongCLIP, our LotCLIP also has better performance**. We implement LotCLIP to fine-tune pre-trained CLIP model using the 1M-scale dataset from ShareGPT4v, similar to LongCLIP.
>
> | Method |Data|Pre-trained CLIP| Long text-image Retrival Avg.|
> |  :----: | :----: | :----: | :----: |
> |  LongCLIP |ShareGPT4V-1M|ViT-B/16|67.16
> |  LotCLIP |ShareGPT4V-1M|ViT-B/16| 82.06 **(+14.9)**
>
> `` Q13: Reason behind the names (Dora and LotCLIP)``
>
> We name the method as **LotCLIP** (**Lo**ng **T**exts in **C**ontrastive **L**anguage-**I**mage **P**re-training) and we name the dataset as **Dora** (**D**etailed Texts **o**f **R**eal Im**a**ges). We will add the explainations in revision.
>
> `` Q14: Explanation on the concept of Corner Tokens``
>
> Sorry for the confusion. We will add the supporting explanations of corner tokens in the updated version. Specifically, we add several learnable tokens from different initialization after [CLS] token, termed as corner tokens.
>
> ``Q15: Training Details``
>
> We apologize for missing the training details, and we add these in table 3 of global pdf. We will fix this issue in the updated version.

---

> ### Author Response · Authors · 2024-08-08
>
> ## Detailed experimental results
>
> Below, we provide detailed experimental results in response to Q2, Q3, Q4, Q5, Q9, Q10, and Q12.
>
>
>
> ``Q2: From an architectural perspective, there seems to be no major difference from [1, 2].``
>
> | Method |  DCI  T2I | DCI I2T| IIW T2I |IIW I2T| SV-10k T2I | SV-10k I2T |  COCO  T2I | COCO  I2T |Avg.|
> |  :----: | :----: | :----: | :----: | :----: | :----: | :----: |:----: |:----: |:----: |
> | Baseline|  47.96 | 44.92 | 84.97 | 81.70 | 73.66 | 66.73  | 30.06 |43.52 | 59.19 |
> | Register |45.62 | 44.65| 83.99 |81.37|74.42| 68.78| 30.34| 44.18|59.17 **(-0.02)** |
> | LotCLIP | 49.46 | 47.82 | 84.97 | 83.33 | 76.49 | 69.72 | 31.59| 46.56 |61.24 **(+2.05)**|
>
> [1] Darcet *et al.*, Vision Transformers Need Registers, in ICLR 2024.
>
> ``Q3: Is there any special reason or evidence that such a technique is especially beneficial for training with long captions?``
>
> | Method| Long Text  | DCI  T2I | DCI I2T| IIW T2I |IIW I2T| SV-10k T2I | SV-10k I2T | COCO  T2I | COCO  I2T | Avg.|
> |  :----: | :----: | :----: | :----: | :----: | :----: | :----: |:----: |:----: |:----: |:----: |
> | LiT| -| 27.14 | 24.13 | 65.20 | 58.50 | 32.73 | 27.01 | 24.07 | 34.20 | 36.62|
> | LotCLIP| - | 27.62 | 25.56 | 63.24 | 58.99 | 32.12 | 27.36 | 24.24| 35.26| 36.80 **(+0.18)**
> | LiT| ✓| 47.96 | 44.92 | 84.97 | 81.70 | 73.66 | 66.73  | 30.06 |43.52 | 59.19 |
> | LotCLIP|✓ |  49.46 | 47.82 | 84.97 | 83.33 | 76.49 | 69.72 | 31.59| 46.56 | 61.24 **(+2.05)**
>
>
> ``Q4: Corner tokens in different position``
>
> | Corner Tokens Position| DCI  T2I | DCI I2T| IIW T2I |IIW I2T| SV-10k T2I | SV-10k I2T | Avg. |
> |  :----: | :----: | :----: | :----: | :----: | :----: | :----: | :----: |
> |  After Text | 45.96 | 46.50 | 85.13 | 81.54 | 77.13 | 70.68 | 67.82|
> |  Before [CLS] | 46.29 | 47.11 | 85.95 | 83.82 | 77.23 | 70.53 | 68.49 **(+0.67)**|
> |  After [CLS] | 49.46 | 47.82 | 84.97 | 83.33 | 76.49 | 69.72 | 68.63 **(+0.81)**|
>
> ``Q5: Corner tokens in vision transformers``
>
>  We are really sorry that the results provided in the response of Q5 are not correct. We implement the corner tokens in the image encoder and find that it provides less improvement (**0.58%** performance improvement) to long-text understanding compared to utilize corner tokens in the text encoder (**1.84%** performance improvement).
>
> | Architecture | Corner Tokens in TE | Corner Tokens in IE | DCI  T2I | DCI I2T| IIW T2I |IIW I2T| SV-10k T2I | SV-10k I2T |  COCO  T2I | COCO  I2T | Avg. |
> |  :----: | :----: | :----: | :----: | :----: | :----: | :----: | :----: | :----: |  :----: |  :----: |   :----: |
> | CLIP |  -| -|   42.92 | 42.23 | 77.29 | 75.98 | 66.78 | 64.70 | 22.33| 32.14 | 53.05
> | CLIP | - | ✓ |  42.79 | 43.05 | 75.98 | 77.12 | 67.97 | 65.62 | 23.03| 33.46 | 53.63 **(+0.58)**
> | CLIP |  ✓ | - | 43.91 | 43.83 | 78.76 | 75.98 | 70.20 | 67.91 | 24.11 |34.40 |54.89 **(+1.84)** |
>
> ``Q9: Verification step for the extracted long captions.``
>
> We utilize GPT4V to assess the alignment between image and text, following Q-Bench [3]. The prompt we used is "{image}. Text: {text}. Please assist in analyzing whether the given text aligns with the given image. Please provide an integer score as a single number from 0 to 5 based on the alignment, without explanation.".
>
> |Source of long text| Score from GPT4v|
> |  :----: | :----: |
> |IIW | 4.51 |
> |InstructBLIP-Vicuna7B | 2.80 |
> |LLaVA-v1.5-13B | 3.45 |
> |ShareGPT4V-13B | 3.48 |
>
> ``Q10: It is unclear whether the comparison is fair. More visual backbones with LiT training.``
>
> |Method| Pretrained Visual Backbone  | DCI  T2I | DCI I2T| IIW T2I |IIW I2T| SV-10k T2I | SV-10k I2T | Avg. |
> |  :----: |  :----: | :----: | :----: | :----: | :----: | :----: | :----: |:----: |
> CLIP| - |11.67 |11.01| 33.17 |31.37| 10.69| 8.77| 17.78 |
> LotCLIP|- |43.91| 43.83 |78.76 |75.98| 70.20 |67.91| 63.43 **(+45.65)** |
> LiT | MoCo-v3 | 20.69| 18.65 | 58.33|52.94 |24.94| 19.31 | 32.48|
> LotCLIP | MoCo-v3 |37.21 | 35.89 | 82.52 | 77.12 | 67.74 | 59.46| 59.99 **(+27.51)**|
> LiT | DINO | 22.45 | 19.42 | 61.11 | 54.74 | 26.53 | 21.40 | 34.28 |
> LotCLIP | DINO | 38.76 | 35.31 | 79.01 | 78.10 | 64.07  | 60.17 | 59.24 **(+24.96)**|
>
>
> ``Q12: Directly comparable to LongCLIP``
>
> | Method |Data|Pre-trained CLIP| DCI  T2I | DCI I2T| IIW T2I |IIW I2T | Share4V-10k T2I | Share4V-10k I2T |  Avg. |
> |  :----: | :----: | :----: | :----: | :----: | :----: | :----: | :----: | :----: | :----: |
> |  LongCLIP |ShareGPT4V-1M|ViT-B/16|47.43 | 44.18 | 89.22 | 86.93  | 73.16 | 62.03| 67.16
> |  LotCLIP |ShareGPT4V-1M|ViT-B/16     |62.74 | 62.96 | 93.30 | 93.46  | 90.42 | 89.47 | 82.06 **(+14.9)**|
>
> ## Looking forward to your feedback
>
> We look forward to hearing from you, and we can further address unclear explanations and remaining concerns, if any.

---

> ### Comment · Reviewer_NJUx · 2024-08-08
>
> I appreciate the authors' extensive efforts in addressing my concerns.
>
> The rebuttal addresses many of the initial issues. In both the main paper and the rebuttal, the proposed framework is effective in tasks related to long texts.
>
> ---
>
> However, my primary concerns on the technical novelty remain unresolved.
> Despite the improved empirical results, the architectural modification on the 'register tokens' seem marginal, particularly in the insertion of learnable tokens into a text encoder instead of a visual encoder.
> Could the authors further elaborate on this point?
>
> Additionally, the modification of the attention mask in LotCLIP appears to offer marginal improvements for corner tokens in Table 3, diminishing its significance as a technical contribution.
> Moreover, the explanation for the improvements in long texts (Q3) is not sufficient for me. Could you share any insights behind these improvements?
>
> In the training dataset and MLLMs, thank you for the further clarifications. I value the substantial volume of the generated data, both in terms of quantity and token lengths.
> However, I perceive the limited technical innovation, since the pipeline simply relies on standard MLLM inference for generating long texts.
>
> I think that the focus of the technical contributions should primarily be on the modeling aspect, which currently seems underdeveloped in this version.
>
> ---
>
> **Further questions** arising from the other reviewers' comments and the authors' responses.
>
> (1) Are the corner tokens defined as separate 'text' elements rather than as learnable token parameters? According to the training process described in the authors' comment, corner tokens undergo the tokenization process alongside the original captions. Could you please clarify this?
>
> (2) Are corner tokens still effective without the BERT's CLS token? For a more comprehensive analysis, I suggest *(e.g., not mandatory)* conducting an experiment, if feasible within the author-reviewer discussion period, that ablates the introduction of LotCLIP, replacing the BERT text encoder with an OpenAI-style text transformer.
> OpenAI counterpart disables the non-causal attention masks, and uses the last token as text pooling.
>
> (3) In measuring long text understanding capabilities with images, the paper considers image-text retrieval. Are there any potential surrogate tasks for assessing understanding in conjunction with long texts and images?

---

> ### Author Response · Authors · 2024-08-09
> **Official Comment by Authors [1/3]**
>
> Thanks for your feedback. We are encouraged by your appreciation on the effectiveness of LotCLIP in tasks related to long texts. And we are glad that our rebuttal addresses some of your concerns. Bellow, we address your remaining concerns and additional questions separately.
>
> ``Q16: However, my primary concerns on the technical novelty remain unresolved. Despite the improved empirical results, the architectural modification on the 'register tokens' seem marginal, particularly in the insertion of learnable tokens into a text encoder instead of a visual encoder. Could the authors further elaborate on this point?``
>
> We are beg to differ with you on this matter.
>
> ***a) Technical difference:***
> Corner tokens and register tokens differ in more than just applying on different encoders. Although the corner tokens and register tokens are both learnable, they differ fundamentally in the following aspects:
>
>
> * **Aligning corner tokens to visual information**: The features of corner tokens are utilized for contrastive learning to align with visual information, while register token outputs are not used for any purpose.
> * **Inserting corner tokens to text dictionary**: The corner tokens are added into the vocabulary of tokenizer and converted to learnable embeddings, as widely used method in the NLP field, *e.g.*, BERT.
> * **Making corner tokens diversity**: Each corner token corresponds to identical token id in the vocabulary of tokenizer ensuring the disparity among corner tokens. Moreover, the interactions between corner tokens and other tokens are restricted by an attention mask mechanism. It promotes the corner tokens to learn diverse text informantion, which helps [CLS] token on image perception.
>
>
>
> ***b) Motivation difference:***
> The corner tokens are designed to assist [CLS] token in **aggregating text token features**, while the register tokens are designed to **mitigate the high-norm outlier tokens** in image patches(tokens). Besides, the high-norm outlier tokens are not salient in language modeling. Thus, the register tokens do not have impacts on long text comprehension.
>
> ***c) Experimental comparision between register and corner tokens:***
>
> **Based on the results of Table 2 and Table 3 of the paper [1], Register tokens may not necessarily make CLIP performance improvment.** It indicates that learnable tokens are not the key to improve short-text understanding of CLIP.
> |  | ImageNet | VOC 2007 |VOC 2012| COCO
> |  :----: | :----: | :----: | :----: | :----: |
> |OpenCLIP|78.2 | 38.8 | 44.3 | 31.0 |
> |OpenCLIP+register token|78.1 **(-0.1)** |37.1 **(-1.7)**|42.0 **(-2.3)**| 27.9 **(-3.1)**|
>
> *All results are derived from the Table 2 and Table 3 of paper [1].
> [1] Vision Transformers Need Registers
>
> **Meanwhile, register tokens are also not the key to improve the long-text understanding of CLIP.** As shown in the table below, we **directly incorperate register tokens into text encoder** on CC3M+short text and CC3M+long text, respectively. It shows incorporating register tokens in the text encoder does not yield improvements in averaged performance on both long-text-image and short-text-image retrieval tasks. Instead, our method boosts baseline by **2.05%** and **0.18%**, respectively, when trained with and without long texts.
>
>
>
> | Long Text | Extra Token | DCI  T2I | DCI I2T| IIW T2I |IIW I2T| SV-10k T2I | SV-10k I2T  | COCO I2T  |COCO T2I  | Avg.|
> |  :----: | :----: | :----: | :----: | :----: | :----: | :----: |:----: |:----: |:----: |:----: |
> | -| - | 27.14 | 24.13 | 65.20 | 58.50 | 32.73 | 27.01 | 24.07 | 34.20 | 36.62|
> | -| register token| 27.37 |24.39|63.73|57.03|32.59|27.11| 24.20|34.98| 36.43 **(-0.19)**|
> | -| corner token | 27.62 | 25.56 | 63.24 | 58.99 | 32.12 | 27.36 | 24.24| 35.26| 36.80 **(+0.18)**
> | ✓| - | 47.96 | 44.92 | 84.97 | 81.70 | 73.66 | 66.73  | 30.06 |43.52 | 59.19 |
> | ✓| register token | 45.62 | 44.65| 83.99 |81.37|74.42| 68.78| 30.34| 44.18|59.17 **(-0.02)** |
> | ✓| corner token| 49.46 | 47.82 | 84.97 | 83.33 | 76.49 | 69.72 | 31.59| 46.56 |61.24 **(+2.05)**|
>
>
>
> ``Q17: Insights behind the improvements in long texts (Q3)``
>
> The corner tokens can facilitate [CLS] token in **aggregating the diverse textual information within long texts** to enhance the long text understanding ability of model. Compared to long texts, short texts have less textual information, where the corner tokens are hard to play a role. Thus, there are less improvement from corner tokens on short texts.

---

> ### Author Response · Authors · 2024-08-09
> **Official Comment by Authors [2/3]**
>
> ``Q18: Limited technical innovation of the generated long texts. The pipeline simply relies on standard MLLM inference for generating long texts.``
>
> Thanks for your valuable suggestion. The Dora dataset is introduced to **fill the need of large-scale long text-image pair dataset in multi-modal learning field**. Existing text-image pair datasets typically consist of short text, restricting the ability of trained models on processing long texts. As far as we known, **Dora is the largest dataset consisting long texts for multi-modal learning**. We believe the community will benefit from Dora dataset in future research. In future, we will use some alleviating hallucination methods (*e.g.*, OPERA [2]) to improve the quality of long captions, which can further improve the dataset.
>
> [2] OPERA: Alleviating Hallucination in Multi-Modal Large Language Models via Over-Trust Penalty and Retrospection-Allocation. 2024.
>
>
> ``Q19: Are the corner tokens defined as separate 'text' elements rather than as learnable token parameters?``
>
> No, the corner tokens are, in fact, learnable token embeddings. In the NLP field, new tokens are typically added to the vocabulary of tokenizer rather than directly initialized as learnable embeddings, *e.g.*, [CLS] and [MASK] token in BERT [3], and "vokens" in MiniGPT-5 [4] and DreamLLM [5].
>
> Concretely, before feeding a text into attention blocks of transformer, a tokenizer is used to convert subwords (text tokens and special tokens, *e.g.*, [CLS] token) into indices, based on their order in the vocabulary of tokenizer [6], *e.g.*, [CLS] token is coverted to 101 by BERT tokenizer. Then, these indices are converted to learnable token embeddings with a lookup table that stores embeddings [6]. In LotCLIP, we extend the vocabulary of tokenizer by adding corner tokens and update the lookup table. In this way, each corner token is converted to a learnable embedding before being fed into attention blocks.
>
> [3] BERT: Pre-training of deep bidirectional transformers for language understanding. 2018.
>
> [4] MiniGPT-5: Interleaved Vision-and-Language Generation via Generative Vokens. 2023.
>
> [5] DreamLLM: Synergistic Multimodal Comprehension and Creation. 2023.
>
> [6] Google's Neural Machine Translation System: Bridging the Gap between Human and Machine Translation. 2016.
>
>
>
> ``Q20: Are corner tokens still effective without the BERT's CLS token?``
>
> If the [CLS] token is removed, corner tokens can also be used to represent texts, because corner token is varient of [CLS] token. As shown in the table below, the performance of using averaged features from corner tokens only degrades using features from the [CLS] token by 0.49% on average.
>
> Method| Text feature |   DCI  T2I | DCI I2T| IIW T2I |IIW I2T| SV-10k T2I | SV-10k I2T  |Avg.|
> |  :----: | :----: | :----: | :----: | :----: | :----: | :----: |:----: |:----: |
> |LiT| feature from [CLS] token  |  47.96 | 44.92 | 84.97 | 81.70 | 73.66 | 66.73 | 66.66
> |LotCLIP| feature from [CLS] token |   49.46 | 47.82 | 84.97 | 83.33 | 76.49 | 69.72 |68.63 **(+1.97)**|
> LotCLIP ([CLS] token is removed)| averaged feature from corner tokens |47.87  |47.41| 84.15|82.19|76.47|70.77 | 68.14 **(+1.48/-0.49)**
>
>
> ``Q21: Ablates the introduction of LotCLIP, replacing the BERT text encoder with an OpenAI-style text transformer``
>
> Thanks for you suggestion. The experimental results of using OpenAI-style text transformer are shown in the Table 4 of the main manuscript, and we also present the results in the following table. CLIP's text encoder disables non-causal attention masks and uses the feature of last token as text feature. For fair comparison, R-LotCLIP adopts the same text encoder architecture. The results demonstrate the effectiveness of LotCLIP with using OpenAI-style text transformer as text encoder.
>
> | Method | Long Text |  DCI  T2I | DCI I2T| IIW T2I |IIW I2T| SV-10k T2I | SV-10k I2T  |Avg.|
> |  :----: | :----: | :----: | :----: | :----: | :----: | :----: |:----: |:----: |
> | CLIP| -|   11.67 |11.01| 33.17 |31.37| 10.69| 8.77| 17.78 |
> | CLIP|✓ | 42.92| 42.23| 77.29| 75.98|66.78|64.70|61.65 **(+43.87)** |
> | R-LotCLIP  |✓ | 43.91| 43.83| 78.76| 75.98| 70.20| 67.91 |63.43 **(+45.65)** |

---

> ### Author Response · Authors · 2024-08-09
> **Official Comment by Authors [3/3]**
>
> ``Q22: Are there any potential surrogate tasks for assessing understanding in conjunction with long texts and images?``
>
> This is an really interesting question. Recent works [7,8], which consider connecting images with long texts, also use long-text-image retrieval to assess the alignment between long texts and images. Beyond retrieval tasks, we believe there are other tasks to assess the benefits from the long text-image alignment, *e.g.*, image understanding with MLLM.
>
> Concretely, we finetune the pre-trained CLIP from openai with the proposed method, where the image and text encoder are unlocked. Then we incorperate the finetuned image encoder with LLM and train the MLLM following LLaVA-1.5. The MLLM is evaluated on two vision-centric VQA benchmarks [9], *i.e.*, MMVP [10] and RealWorldQA [11], as shown in following table. The results indicate that using long texts for constrastive learning enhances the image encoder's ability to extract accurate and comprehensive visual features, which leading to improved image understanding by Multimodal Language Models (MLLMs).
>
> | Image Encoder | MMVP |  RealWorldQA |
> |  :----: | :----: | :----: |
> |OpenAI CLIP| 19.3|50.3
> |LotCLIP| 26.7 **(+7.4)**| 51.9 **(+1.6)**
>
> [7] Long-CLIP: Unlocking the Long-Text Capability of CLIP. 2024.
>
> [8] MATE: Meet At The Embedding -- Connecting Images with Long Texts. 2024.
>
> [9] Cambrian-1: A Fully Open, Vision-CentricExploration of Multimodal LLMs. 2024.
>
> [10] Eyes Wide Shut? Exploring the Visual Shortcomings of Multimodal LLMs. 2024.
>
> [11] Grok-1.5 Vision Preview. 2024.

---

> ### Comment · Reviewer_NJUx · 2024-08-11
>
> From the additional clarifications and experiments, the proposed corner token for the text encoder appears to be both innovative and more effective than the register token scheme used in the visual encoder, particularly in processing long captions. I believe the authors' presentation in the rebuttal has convincingly demonstrated its strength, leaving me with no remaining questions. Therefore, I no longer support rejecting it (score: 3 to 5).
>
> Meanwhile, I view the 'register token' scheme as a crucial baseline for emphasizing the novelty and superiority of the proposed corner token, since both introduce additional learnable tokens into the pre-trained model. However, because the discussion about the register token was omitted in the initial draft, I am concerned that incorporating all discussions related to the register token, which have so far been addressed in the rebuttal, might require a significant revision.

---

> > ### Author Response · Authors · 2024-08-11
> >
> > We will include the discussions related to the register token in the revision. Thank you again for your thoughtful reviews and discussions, which have greatly elevated the quality of our paper!

---

### Official Review · Reviewer_sjvn · 2024-07-11

**Soundness:** 3
**Presentation:** 3
**Contribution:** 3
**Rating:** 7
**Confidence:** 4

**Summary:**

To improve the ability of vision-language models (VLMs) for long-text understanding, the paper proposes to relabel the data with long captions, however, direct learning may lead to performance degradation in understanding short text (e.g., in the image classification task). Then, corner tokens are introduced to aggregate diverse textual information, enabling the model to catch up to its original level of short-text understanding yet greatly enhance its capability of long-text understanding. Experiments are performed on a large-scale long caption dataset to demonstrate the effectiveness of the proposed method.

**Strengths:**

1) The author points out the phenomenon that the key reason causing such an issue is that the training images are usually paired with short captions, leaving certain tokens easily overshadowed by salient tokens.
2) To improve the ability of vision-language models (VLMs) for long-text understanding, the paper proposes to relabel the data with long captions; corner tokens are introduced to aggregate diverse textual information, enabling the model to catch up to its original level of short-text understanding yet greatly enhance its capability of long-text understanding.
3) Experiments are performed on a large-scale long caption dataset to demonstrate the effectiveness of the proposed method.

**Weaknesses:**

1) Lack of details for addressing the limitation for the token length limitation of the text encoder in Sec. 3.4.
2) For the attention mask part, if the corner tokens can be seen by other text tokens, how does it perform? In Table 3, how the model will the model perform if the corner taken is removed?
3) The year of references for NeurIPS papers e.g., [10] and [14], is not correct.

**Questions:**

Please refer to the weakness part.

**Limitations:**

Yes. Both the limitation and potential impact are well described in the paper.

---

> ### Author Rebuttal · Authors · 2024-08-07
>
> Thank you for your positive comments and valuable feedback on our work! We are excited and encouraged by your support! Bellow we address your concern separately.
>
> ``Q1: Lack of details for addressing the limitation for the token length limitation of the text encoder in Sec. 3.4.``
>
> Sorry for confusion. We set a token length limitation of 128 for the text encoder based on the following two considerations:
> - **Token length of training text.** For pre-trained models, the token length limitation can be arbitrarily set to any positive integer value. As shown in Figure 4 of Sec. 3.4, the results indicate that a text input limitation of 77 tokens is insufficient for model that requires long text comprehension. However, the training dataset has average of 136 tokens per text, and larger token length limitation may not bring more information.
> - **Balance of training efficiency and performance.**: In Figure 4 of Sec. 3.4, a smaller token number may lead to performance degradation due to insufficient encoding of text information, while a larger token number may increase computational complexity. Thus, to balance the training efficiency and performance, we generally choose 128 as the maximum token length.
>
> We will add more details in updated version.
>
> ``Q2: For the attention mask part, if the corner tokens can be seen by other text tokens, how does it perform? In Table 3, how the model will the model perform if the corner taken is removed?``
>
>
> Thanks for the valuable suggestion. We have added the following experiments:
>
> a) **Performance is not improved (68.19 *v.s.* 68.20) when the corner tokens can be seen by other text tokens.** This is because allowing corner tokens to be seen by text tokens can influence the diversity of aggregated text features, leading to worse performance.
>
> b) **Removing the corner token degrades LotCLIP's average performance by 1.98%**, which further demonstrates the effectiveness of corner tokens.
>
> We will include the results in Table 3 of the revision.
>
>
> |  Corner Token |  Attention Mask Mechanism | DCI  T2I | DCI I2T| IIW T2I |IIW I2T| SV-10k T2I | SV-10k I2T | Avg. |
> |  :----: | :----: | :----: | :----: | :----: | :----: | :----: | :----: |:----: |
> |-| - | 47.96 | 44.92 | 84.97 | 81.70 | 73.66 | 66.73  | 66.65
> |✓| - | 48.61 | 47.17 | 86.11 | 81.86 | 76.14 |  69.31|   68.20 **(+1.55%)**
> |✓| text tokens can see corner token | 48.29 | 47.23 | 85.29 | 82.26 | 76.70| 69.38| 68.19 **(+1.54%)**
> |✓| text tokens can't see corner token | 49.46 | 47.82 | 84.97 | 83.33 | 76.49 | 69.72 | 68.63 **(+1.98%)**
>
>
> ``Q3: The year of references for NeurIPS papers e.g., [10] and [14], is not correct.``
>
>
> We apologize for any previous errors in the references, which we will carefully verified and rectified in the revised paper.
>
> **Incorrect Reference in paper**:
>
> [10] L. Fan, D. Krishnan, P. Isola, D. Katabi, and Y. Tian. Improving clip training with language rewrites. Advances in Neural Information Processing Systems, 36, 2024
>
> [14] J. Lee, J. Kim, H. Shon, B. Kim, S. H. Kim, H. Lee, and J. Kim. Uniclip: Unified framework for contrastive language-image pre-training. Advances in Neural Information Processing Systems, 35:1008–1019, 2022
>
> [25] Y. Tian, L. Fan, P. Isola, H. Chang, and D. Krishnan. Stablerep: Synthetic images from text-to-image models make strong visual representation learners. Advances in Neural Information Processing Systems, 36, 2024.
>
> **Rectification**:
>
> [10] L. Fan, D. Krishnan, P. Isola, D. Katabi, and Y. Tian. Improving clip training with language rewrites. Adv. Neural Inform. Process. Syst., **36:35544–35575, 2023**
>
> [14] J. Lee, J. Kim, H. Shon, B. Kim, S. H. Kim, H. Lee, and J. Kim. **UniCLIP**: Unified framework for contrastive language-image pre-training. Adv. Neural Inform. Process. Syst., 35:1008–1019, 2022
>
> [25] Y. Tian, L. Fan, P. Isola, H. Chang, and D. Krishnan. StableRep: Synthetic images from text-to-image models make strong visual representation learners. Adv. Neural Inform. Process. Syst., **36:48382–48402, 2023**

---

> > ### Comment · Reviewer_sjvn · 2024-08-12
> > **Official Comment by Reviewer sjvn**
> >
> > Thanks for the response from the authors.
> >
> > My concerns are well solved in the rebuttal. After considering other reviews and the corresponding answers, I'd like to keep the rating at the current stage.

---

> > > ### Author Response · Authors · 2024-08-12
> > >
> > > Thanks for your feedback, we are glad that our response addresses your concerns. Thank you again for your thoughtful reviews, which have greatly elevated the quality of our paper!

---

### Official Review · Reviewer_Zcuk · 2024-07-12

**Soundness:** 2
**Presentation:** 2
**Contribution:** 2
**Rating:** 5
**Confidence:** 4

**Summary:**

The paper addresses a significant gap in current language-image pre-training models, which are typically trained on datasets with short captions. This limitation hinders the models' ability to effectively understand and process long texts. The proposed solution, LotCLIP, introduces methods to enhance long-text understanding without compromising the performance on short-text tasks.

**Strengths:**

(a) The authors re-captioned 100 million images with long texts using multi-modality large language models.

(b) Introducing the concept of  corner token

**Weaknesses:**

(a) Not very clear why the proposed method is based on CLIP architecture only- could it have been built upon other similar algorithm as well, for example – ALIGN ?

(b) Need to illustrate in detail how this “corner token” learning is actually taking place.

(c) Should have also reported results on  ALIGN


(d) The references are not correct/incomplete in some occasions -please cross check all.

**Questions:**

Could you please provide a detail diagram on training process encompassing the "corner tokens ?

**Limitations:**

No limitations were mentioned by the authors.

***** Raising the final score to 5  from 4********************

---

> ### Author Rebuttal · Authors · 2024-08-07
>
> Thank you for your positive comments and valuable feedback on our work! We are excited and encouraged by your support! Bellow we address your concern separately.
>
> ``Q1: LotCLIP on other similar algorithms, *e.g.* ALIGN, CoCa.``
>
> **Our LotCLIP can be applied on other similar algorithms as well, *e.g.* ALIGN, CoCa.** The results on CC3M are presented in the table below, which demonstrate that LotCLIP also enhances the long text comprhension ability of other similar algorithm besides CLIP.
>
> [1] CoCa: Contrastive Captioners are Image-Text Foundation Models
>
> |  Algorithm   | DCI  T2I | DCI I2T| IIW T2I |IIW I2T| SV-10k T2I | SV-10k I2T | Avg
> |  :----: |  :----: | :----: | :----: | :----: | :----: | :----: |:----: |
> | ALIGN | 19.67 | 18.01 | 52.45 | 50.16 | 24.21 | 18.33 | 30.47|
> | +Long Caption | 39.68 | 38.48 | 79.58 | 76.31| 63.53 | 39.68| 56.21 **(+25.74)** |
> | +Coner Token |  40.49 | 40.87 | 81.70 | 78.76 | 69.94 | 67.50 | 63.21 **(+32.74)**|
> | CoCa | 9.04 | 8.66 | 27.45 | 27.94 | 9.46 | 9.12 | 15.28|
> | +Long Caption |32.47 | 31.21 |66.17|65.68|54.10|51.98|50.27 **(+34.99)**|
> | +Coner Token |  34.49 | 33.14 | 68.95 | 67.32 | 58.64 | 56.01 | 53.09 **(+37.81)**|
>
> ``Q2: Illustrate learning process of corner tokens``
>
> Thanks for the valuable suggestion. The corner tokens are learnable tokens, which are placed before text tokens and after [CLS] token. We demonstrate the training process of corner tokens in below.
>
> **Inputs**:
> - image\_encoder
> - text\_encoder
> - tokenizer
> - image\_processor
> - learnable corner tokens: $[\texttt{Cor 1}], [\texttt{Cor 2}],\cdots, [\texttt{Cor m}]$
> - minibatch of long texts: $T_{\text{long}}$ [$n,l$]
> - minibatch of short texts: $T_{\text{short}}$ [$n,l$]
> - minibatch of images: $I$ [$n,h,w,c$]
> - learned temperature parameter: $t$
>
> **Training Process**:
>
> *\#* The corner tokens are placed in front of text before the tokenization process.
>
> *\#* After tokenization, the first token of text input is [CLS] token,
>
> *\#* while the i-th token (1<i<=$m$+1) is $[\texttt{Cor i-1}]$ token.
>
> $T_{\text{long}}$ $=$ tokenizer($[\texttt{Cor 1}], [\texttt{Cor 2}],\cdots, [\texttt{Cor m}]$+ $T_{\text{long}}$)
>
> $T_{\text{short}}$ $=$ tokenizer($[\texttt{Cor 1}], [\texttt{Cor 2}],\cdots, [\texttt{Cor m}]$ + $T_{\text{short}}$)
>
> *\#* Image pre-processing
> $I$ $=$ image\_processor($I$)
>
> *\#* Build attention mask mechanism
> $\mathcal{A}$ $=$ np.ones([$l$,$l$])
>
> *\#* Text tokens do not attend to corner tokens
> $\mathcal{A}[m+1:, 1:m+1]$ $*=$ 0
>
> *\#* [CLS] token and corner tokens do not interact with each other
> $\mathcal{A}[:m+1, :m+1]$ $=$ np.eye(m+1)
>
> *\#* The attention mask $\mathcal{A}$ is multiplied by original attention mask used in the text encoder
>
> *\#* to eliminate the influence from [PAD] tokens to [CLS] and text tokens.
>
> $\mathcal{A_{\text{long}}}$ $=$ $\mathcal{A}$ $\cdot$ text\_encoder.build_attn_mask($T_{\text{long}}$)
>
> $\mathcal{A_{\text{short}}}$ $=$ $\mathcal{A}$ $\cdot$ text\_encoder.build_attn_mask($T_{\text{short}}$)
>
> *\#* Extract text features.
>
> *\#* The attention mask controls the interaction among tokens within in each attention block.
>
> *\#* The text\_encoder outputs the features of the first $m+1$ tokens, where the first
>
> *\#* one is [CLS] token and the remains are corner tokens.
>
> $f_{lt}$ $=$ text\_encoder($T_{\text{long}}$, attention\_mask $=$  $\mathcal{A_{\text{long}}}$) *\#* [$n,m+1,d$]
>
> $f_{st}$ $=$ text\_encoder($T_{\text{short}}$, attention\_mask $=\mathcal{A_{\text{short}}}$) *\#* [$n,m+1,d$]
>
> *\#* Extract image features.
>
> $f_\text{i}$ $=$ image\_encoder($I$) *\#* [$n,d$]
>
> *\#* Normalization
>
> $f_{lt}$ $=$ l2\_normalize($f_{lt}$, axis=-1)
>
> $f_{st}$ $=$ l2\_normalize($f_{st}$, axis=-1)
>
> $f_{i}$ $=$ l2\_normalize($f_{i}$, axis=-1)
>
> labels $=$ np.arange(n)
>
> *\#* Loss computation
>
> $logits_{\text{short}}$ $=$np.dot($f_{i}$, $f_{st}$[:,0,:]) $\cdot$ np.exp($t$)
>
> $loss_{\text{short}}^{i2t}$ $=$ cross\_entropy\_loss($logits_{\text{short}}$, labels, axis=0)
>
> $loss_{\text{short}}^{t2i}$  $=$ cross\_entropy\_loss($logits_{\text{short}}$, labels, axis=1)
>
> $loss_{\text{short}}$ $=$ ($loss_{\text{short}}^{i2t}$+$loss_{\text{short}}^{t2i}$)/2
>
> $loss_{\text{long}}$ $=$ 0
>
> For $k \in [0, m]$:
>
> &nbsp;&nbsp;&nbsp;&nbsp;&nbsp; $logits_{\text{long}\_k}$ $=$ np.dot($f_\text{image}$, $f_\text{long}$[:,$k$,:]) * np.exp($t$)
>
> &nbsp;&nbsp;&nbsp;&nbsp;&nbsp; $loss_{\text{long}\_k}^{i2t}$ $=$ cross\_entropy\_loss($logits_{\text{long}\_k}$, labels, axis=0)
>
> &nbsp;&nbsp;&nbsp;&nbsp;&nbsp; $loss_{\text{long}\_k}^{t2i}$ $=$ cross\_entropy\_loss($logits_{\text{long}\_k}$, labels, axis=1)
>
> &nbsp;&nbsp;&nbsp;&nbsp;&nbsp; $loss_{\text{long}}$ $+=$ ($loss_{\text{long}\_k}^{i2t}$+$loss_{\text{long}\_k}^{t2i}$)/2
>
> loss $=$ $loss_\text{short}$ + $loss_{\text{long}}$
>
> ``Q3: Should have also reported results on ALIGN``
>
> Thanks for your valuable suggestion. **We report the performance of ALIGN** pre-trained on COYO-700M in the table below. Even trained with smaller scale dataset, LotCLIP outperforms ALIGN on long-text-image retrieval task. We will include the results of ALIGN for comparision in the revised version of our paper.
>
> |  Method   | Data Scale   | DCI  T2I | DCI I2T| IIW T2I |IIW I2T| SV-10k T2I | SV-10k I2T | Avg. |
> |  :----: | :----: | :----: | :----: | :----: | :----: | :----: | :----: | :----: |
> |  ALIGN | COYO-700M |56.54 | 57.41 | 92.65 | 90.68 | 65.13 | 62.73| 70.86 |
> |  LotCLIP | Dora-100M | 62.10 | 61.06 | 93.95 | 92.48 | 86.84 | 81.40 | 79.64 **(+8.78)** |
>
> |  Method   | Data Scale   | ImageNet CLS | COCO I2T| COCO T2I | Avg. |
> |  :----: | :----: | :----: | :----: | :----: | :----: |
> |  ALIGN | COYO-700M |65.89 |  60.42 | 42.36 |56.22|
> |  LotCLIP | Dora-100M | 72.16 | 59.66 | 38.06 |56.62 **(+0.40)** |
>
> ``Q4: Not correct/incomplete references``
>
> We apologize for any previous errors in the references, which we will carefully verify and rectify in the revised paper.
>
> ``Q5: Diagram on corner token learning process``
>
> Please refer to our reply to Q2.

---

> > ### Comment · Reviewer_Zcuk · 2024-08-12
> > **Response to the Authors**
> >
> > Dear Authors,
> >
> > Thanks for your detail explanation. I am satisfied.  I have no further questions.
> >
> > With Regards,
> >
> > Zcuk

---

> > > ### Author Response · Authors · 2024-08-12
> > > **Response to Reviewer Zcuk**
> > >
> > > We sincerely appreciate your valuable feedback and are pleased to hear that our response addresses your concerns. We will revise the manuscript as suggested. If you have any further concerns, please feel free to let us know.

---

### Official Review · Reviewer_f34j · 2024-07-12

**Soundness:** 3
**Presentation:** 3
**Contribution:** 3
**Rating:** 5
**Confidence:** 3

**Summary:**

The paper describes a framework to adapt language-image pre-training models to longer captions. For that, first a new dataset is created with longer captions and second, training is modified to adapt to longer captions. New corner tokens are introduced that are supposed to capture longer dependencies in the text, and the training loss is modified to take into account both short and long text. Experimental results show that the method performs better than other methods for tasks including both long-text retrieval and short-text retrieval.

**Strengths:**

- A new dataset is created that can be used in the future for improving the ability of the models for long-text pairing with images
- Several modifications are included in the training procedure that allow to take into account long text while preserving the performance with short text. These modification include modifying the text representation with the corner tokens, introducing a specific attention mask and defining a new loss balancing loss for long and short text.
- Experiments show that the proposed method performs better than other existing approaches both in long text and short text retrieval. Experiments includes a detailed ablation study analyzing the impact of the different components of the model.

**Weaknesses:**

It is not clear the difference of the new Dora dataset with the dataset proposed in DreamLIP (reference [34]) where 30M images are also re-captioned with MLLMs. More details on how the Dora dataset has been built will be useful and also some statistics with respect of the length of the captions in the dataset, comparing with the other datasets analyzed in table 1

**Questions:**

See above in weaknesses

**Limitations:**

There is no specific discussion on the limitations of the method

---

> ### Author Rebuttal · Authors · 2024-08-07
>
> Thank you for your positive comments and valuable feedback on our work! We are excited and encouraged by your support! Bellow we address your concern separately.
>
> ``Q1: Difference between Dora dataset and DreamLIP.``
>
>
> The main difference between Dora dataset and the dataset proposed in DreamLIP lies on two aspects:
> - **Larger Dataset Scale**: Our dataset has 8x image volume and 5x text volume of DreamLIP as shown in following table.
> - **Longer Token Length**: Our dataset has average of 136 tokens per text, which is twice as many as DreamLIP (136 *v.s.* 75).
>
> It is mentioned that the long texts of DreamLIP are splitted into short texts and sperately utilized. In this way, the usage text of DreamLIP are tokenized to around **21 tokens** on average, which not exploring how to make text encoders better understand long texts.
>
>
> | Dataset   | Num. of Image  |  Num. of Long Caption  |  Tokens per Long Text  |
> | :----: | :----: | :----: | :----: |
> | DreamLIP-15M [1] |  13,464,144| 80,784,864 |  74.86 |
> | **Dora**  | **102,571,723** | **307,715,169** | **136.14** |
>
> [1] We report the statistical results derived from 15 million data released by DreamLIP.
>
> `` Q2: More details on how the Dora dataset has been built will be useful.``
>
> Thanks for your valuable suggestion. We use multi-modal large language models (MLLMs) to generate long texts by prompting 'Please describe the image in details'. To prevent the bias introduced by MLLMs' hallucinations, three kinds of MLLMs are introduced to generate the diverse long texts from different model knowledges. We also provide the hyper-parameter settings of the used MLLMs:
> |Hyper-parameters|ShareGPT4V-13B | LLaVA-v1.5-13B |InstructBLIP-Vicuna7b|
> |  :----: | :----: | :----: | :----: |
> |max_new_tokens | 1024 | 512 |  256 |
> |num_beams | 5 |1|5|
> |do_sample| True |True|False|
> |top_p | None|None|0.9|
> |temperature|0.2|0.2|1|
>
>
> `` Q3: Statistics of Dora dataset, comparing with the other datasets analyzed in table 1 of the main manuscript.``
>
> As shown in the table below, we report some statistics of our dataset. To the best of our knowledge, **Dora is the largest dataset consisting long texts for multi-modal learning**. And we are continuing expanding the size of Dora by integrating additional MLLMs for long text generation, as well as gathering more publicly available datasets.
>
>
> | Dataset   | # Num. of Image  |  # Num. of Long Caption  |  # Tokens per Long Text  |
> | :----: | :----: | :----: | :----: |
> | MSCOCO  | 5,000|25,000|11.77|
> | Flickr30k  | 1,000 | 5,000|14.03|
> | DCI  | 7,805 | 7,805 | 172.73 |  172.73 |
> | IIW  | 612| 612 |239.73 |
> | ShareGPT4v-10k  |  10,000|10,000|  173.66 |
> | DreamLIP-15M  |  13,464,144| 80,784,864 | 74.86 |
> | **Dora**  | **102,571,723** | **307,715,169** | **136.14** |

---

> > ### Comment · Reviewer_f34j · 2024-08-13
> >
> > I want to thank the authors for their clarification on the dataset. I have no further questions

---

> > > ### Author Response · Authors · 2024-08-13
> > > **Official Comment by Authors**
> > >
> > > Thanks for your feedback. We are glad that our response addresses your concerns. We will include more details about the dataset in the revision.

---

### Author Rebuttal · Authors · 2024-08-07

Dear reviewers,

We thank all reviewers for their time and efforts in reviewing our paper. These constructive reviews can bring the improvements for our manuscript. We are encouraged that the reviewers appreciate our method, including

* Problem definition and analysis (Reviewer sjvn, NJUx)
* Effective method (Reviewer f34j, sjvn)
* Strong and detailed experiments (Reviewer f34j, NJUx)
* Valuable dataset (Reviewer f34j, Zcuk, NJUx)

We also have made diligent efforts to address all the concerns raised point by point. In this rebuttal, we have incorporated some new figures to more effectively address the concerns. Kindly review the newly uploaded one-page PDF.

* Table 1 gives statistics about the training data (Reviewer NJUx, Q6)
* Table 2 gives hyper-parameter settings of the used MLLMs.(Reviewer NJUx, Q7)
* Table 3 gives training hyper-parameters of our model (Reviewer NJUx, Q15)

We are open to discussions and addressing any issues from reviewers. Your constructive comments can further help us to improve our method.

Sincerely yours,

Author

---

### Decision · Program_Chairs · 2024-09-25

**Decision:**

Accept (poster)

**Comment:**

The manuscript received positive reviews from all reviewers. The authors response further clarified aspects raised by the reviewers, while a lot of detailed information was presented by the authors.

I consider the paper worth presenting in the conference, and encourage the authors to properly integrate new information from the reviewing process in the final revision.